# T-MARS : Improving Visual Representations by Circumventing Text Feature Learning

**Pratyush Maini**[*†]  **Sachin Goyal**[*†]  **Zachary C. Lipton**[†]  **J. Zico Kolter**[†‡]  **Aditi Raghunathan**[†]

Carnegie Mellon University[†]    Bosch Center for AI[‡]

{pratyushmaini,sachingoyal,zlipton,zkolter,raditi}@cmu.edu

## Abstract

Large web-crawled multimodal datasets have powered a slew of new methods for learning general-purpose visual representations, advancing the state of the art in computer vision and revolutionizing zero- and few-shot recognition. One crucial decision facing practitioners is how, if at all, to curate these ever-larger datasets. For example, the creators of the LAION-5B dataset chose to retain only image-caption pairs whose CLIP similarity score exceeded a designated threshold. In this paper, we propose a new state-of-the-art data filtering approach motivated by our observation that nearly $40\%$ of LAION's images contain text that overlaps significantly with the caption. Intuitively, such data could be wasteful as it incentivizes models to perform optical character recognition rather than learning visual features. However, naively removing all such data could also be wasteful, as it throws away images that contain visual features (in addition to overlapping text). Our simple and scalable approach, T-MARS (Text Masking and Re-Scoring), filters out only those pairs where the text dominates the remaining visual features—by first masking out the text and then filtering out those with a low CLIP similarity score of the masked image with original captions. Experimentally, T-MARS is the top ranked approach on Imagenet at "medium scale" of DataComp (a data filtering benchmark), and outperforms CLIP filtering by a margin of $6.5\%$ on ImageNet and $4.7\%$ on VTAB. Additionally, we show that the accuracy gains enjoyed by T-MARS linearly increase as data and compute are scaled exponentially.

## 1 Introduction

The paradigm of machine learning has shifted from training on carefully crafted labeled datasets to training on large crawls of the web (Bommasani et al., 2021). Vision-language models like CLIP (Radford et al., 2021) and BASIC (Pham et al., 2021) trained on web-scale datasets have demonstrated exceptional zero-shot performance across a wide range of vision tasks, and the representations that they learn have become the de facto standard across a variety of vision domains. Recently, the Open-CLIP (Ilharco et al., 2021) effort has aimed to independently reproduce the performance of the original CLIP model through the curation of a similarly sized LAION-400M (Schuhmann et al., 2021) dataset. However, they are still unable to match the performance of CLIP, suggesting that data curation could play an important role even at web-scale. Most recently, the launch of 'DataComp' (Gadre et al., 2023), a data filtering competition at various web-scale, has further streamlined efforts in this field.

Data curation at web scale raises unique challenges compared to the standard classification regime. In web-scale datasets, we typically make only a single (or few) pass(es) over each training example (Hoffmann et al., 2022), as it is often beneficial to see a fresh batch of data from the virtually unbounded web-scale data. However, prior data pruning approaches that characterize the hardness of individual data points (Mindermann et al., 2022; Maini et al., 2022) were proposed for, and evaluated on models trained to convergence in the standard setting. More importantly, any data curation method has to be adaptable to the multimodal contrastive learning setting, and scalable to billions of samples, rendering several prior methods simply infeasible (Sorscher et al., 2023).

---

[*] Equal Contribution.

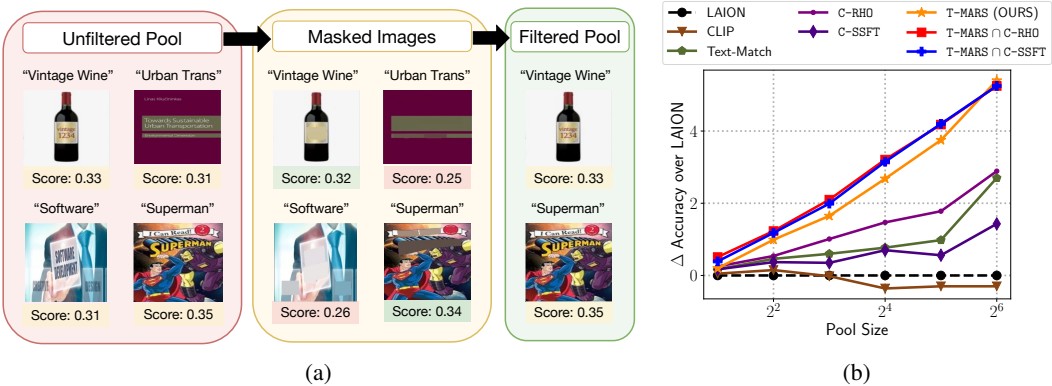

(a)  (b)

Figure 1: (a) Given an unfiltered pool of image-caption pairs, `T-MARS` first masks the text present in each image, and calculates the similarity between the masked image and the corresponding caption, retaining only those with high similarity scores. (b) Scaling curves depicting a linear increase accuracy as data is increased exponentially when training ViT-B-32 models on filtered data versus training on the LAION dataset. The training compute is scaled proportionally with the pool size.

In this work, we propose a new state-of-the-art data filtering approach for large-scale image-text datasets. We start by looking at how the image and text modalities interact in these datasets. We find that around $40\%$ of examples in the LAION dataset have text in the image—for example book covers (Figure 1). This text is often the only element correlated with the caption, necessitating that the model learns to solve an "optical character recognition" (OCR) task in order to minimize the contrastive loss. This is wasteful if we were only interested in purely visual features which are relevant for downstream vision tasks. Conversely, however, naively removing *all* such images that contain text (e.g., similar to Radenovic et al. (2023)), discards a substantial portion of images that contain both visual and well as text features. For example, the "vintage wine" image from Figure 1 provides useful visual cues about what a bottle of wine looks like, despite containing overlapping text with caption.

Our simple and scalable method, Text-Masking and Re-Scoring (`T-MARS`) filters out examples where the text feature dominates the visual features in their contribution to matching the corresponding caption. Specifically, we first mask the text inside the images and then calculate the cosine similarity score of the masked image embedding with that of the caption. Finally, we filter out images with a low similarity score (see Figure 1a). We establish `T-MARS` as a state-of-the-art data filtering technique, by extensively evaluating on 6 different subsets of LAION at exponentially increasing scales (2M to 64M), where `T-MARS` outperforms the most competitive baseline by as much as $3.7\%$ on ImageNet zero-shot accuracy. On a recently released data-filtering benchmark DataComp (Gadre et al., 2023), `T-MARS` is currently the top-ranked approach on Imagenet at 'medium scale' and outperforms CLIP filtering by more than $6.5\%$. We additionally present *scaling experiments* for our approach: through experiments on pool sizes ranging from 2M to 64M, we showcase a surprising linear increase in accuracy gains as the pool size is scaled exponentially (Figure 1). Our scaling trends show that good-quality data filtering holds even more significance at large scales.

To develop a fundamental understanding behind our gains, we plot *utility curves* for various image types (based on the features present) by modifying the ImageNet-Captions dataset (Fang et al., 2022). Our experiments show that (a) images with both visual and text features have nearly the same utility as those with just visual features; and (b) images with only text have the same negative utility as mislabeled samples (§ 6), hurting downstream performance. Finally, we also introduce and benchmark two competitive data filtering baselines, `C-RHO` and `C-SSFT`, by drawing insights from the long line of work in the supervised classification regime on hard example mining (§ 4.2). These baselines themselves perform better than widely used CLIP score-based filtering and notably can boost `T-MARS` performance when we take an intersection of the data retained by `T-MARS` and the proposed baselines.

With the ML community focused on scaling up dataset sizes, our experiments most notably show that pruning off 'bad data' can have $3\times$ more utility than adding more 'good' samples to the dataset.

## 2    RELATED WORK

**Data Curation for Web-Scale Datasets**    Following the curation of the LAION-5B (Schuhmann et al., 2021; 2022) datasets, there has been a growing interest in exploring improved strategies for selecting subsets of the common crawl that help learn better visual representations. Radenovic et al. (2023) suggested using a mixture of three metrics, namely, complexity, action, and text-match (Does the associated caption describe an action that contains a complex object relationship? Does the text in the image match with a part of the caption?). Retaining examples based on complexity and action metrics is seen to hurt zero-shot performance, whereas filtering out examples with text-match helps. This work required text-recognition to match the text with caption, which requires an order of magnitude more compute than text detection required for our proposed masking approach. Recently, similar to the use of synthetic captions in training image captioning models (Li et al., 2023), Nguyen et al. (2023) proposed generating new captions for web-crawled images using an off-the-shelf image captioning model. They filter data points based on CLIP score of the images and synthetic captions.

Abbas et al. (2023) noted that web-scale datasets have a large number of near and exact duplicates, and removed such duplicates to speed up training. CiT (Xu et al., 2023) proposed to select relevant samples based on the match between captions and metadata (ex. class names) of downstream tasks. However, this method does not allow learning general-purpose vision representations. Recently, DataComp (Gadre et al., 2023) was introduced as a benchmark challenge for subset selection from the common crawl. Filtering data based on CLIP score was the best-performing baseline approach.

**Hard Example Mining in Supervised Classification**    In the image classification paradigm, multiple works have focused on finding and prioritizing training on hard examples, which are filtered using memorization and influence scores (Feldman & Zhang, 2020; Feldman, 2020; Jiang et al., 2020), or based on the learning dynamics of different samples (Chatterjee, 2020; Mangalam & Prabhu, 2019; Shah et al., 2020; Kaplun et al., 2022; Carlini et al., 2019). More recent works studying realistic dataset settings such as those with noisy examples discovered that prioritizing so-called 'hard' examples may be a suboptimal approach because it also incentivizes prioritizing the training on mislabeled examples. Mindermann et al. (2022) proposed the RHO (robust hold-out) loss and Maini et al. (2022) proposed the SSFT (second-split forgetting time) towards identifying mislabeled examples. In Section 4.2, we discuss our adaptations of these ideas in the contrastive loss setting.

**Vision-language pre-training**    Image-language contrastive pre-training on web-scale datasets has gathered significant interest from the research community, because of the impressive zero-shot performance on the downstream tasks (Radford et al., 2021; Ilharco et al., 2021; Yao et al., 2021; Goyal et al., 2022; Mu et al., 2021; Li et al., 2022). CLIP (Radford et al., 2021) released the first large-scale vision-language model, obtaining around $75\%$ zero-shot accuracy on ImageNet. BASIC (Pham et al., 2021) scaled up the model size, compute, and data to further drive up performance gains. In this work, we aim to improve the zero-shot performance by *only* modifying the subset of data we train on.

## 3    WHAT CONSTITUTES THE LAION DATASET? A PILOT STUDY

An analysis of image-caption pairs in web-crawled datasets is crucial to understanding the features in the image that models may utilize to align image and caption embeddings. To address this, we perform a small pilot study on 500 image-caption pairs from the LAION dataset (see Appendix D.2 for why 500 samples can be a representative subset). Our analysis yields an interesting observation— approximately 40% of the images possess "text" features (i.e. text written on the image) that correlate with the caption. In fact, nearly 20% times such text features constitute the *sole element* in the image that is correlated with the caption (eg. Book Covers). However, at the same time, a substantial fraction of these images exhibit both text features and general visual cues. For example, an image of a wine bottle with the word "vintage" written on it, accompanied by the caption "vintage wine". These observations lead us to classify the data into five categories based on the correlation between image features (text or visual) and the caption (See Figure 2):

1. *Un-correlated Image and Caption* ($\mathcal{S}_r$; 3.7%): These pairs are essentially mislabeled, with no correlation between the image and caption. These typically exist due to missing image links.

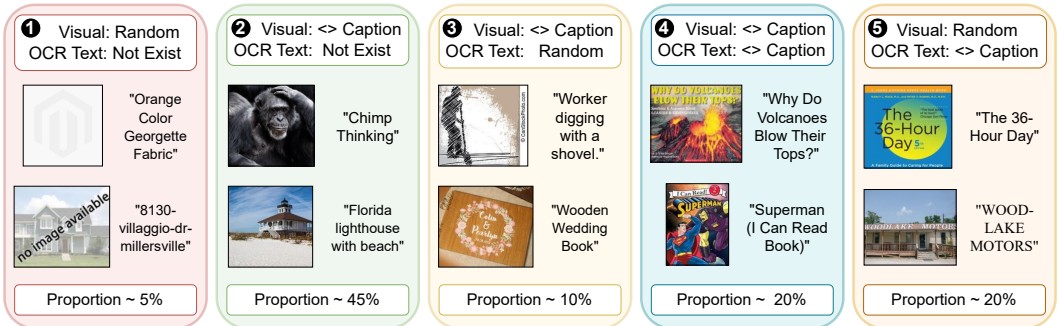

Figure 2: A representation of the various *types* of examples in the LAION dataset. '<>' reads as 'is correlated with'. A significant proportion of examples have some form of text overlayed on the image.

2. *Correlated Visual Feature and Caption* ($\mathcal{S}_i$; $46.7\%$): This is the most common category, where the image accurately corresponds to the caption and contains no text.

3. *Correlated Visual Feature and Caption, Random OCR Text* ($\mathcal{S}_{irt}$; $9.8\%$): Some images include unrelated random text, such as website names. The model would typically disregard such text as it does not contribute to aligning the embedding with the caption.

4. *Both Visual Feature and OCR Text correlated with Caption* ($\mathcal{S}_{it}$; $19.1\%$): These images contain both text and visual features that are correlated with the caption. For instance, in category 4 of Figure 2, the image of Superman includes the visual representation of Superman as well as the text "I can read: Superman," which aligns with the caption. It remains unclear whether the model would prioritize text or visual features in such cases.

5. *Correlated OCR Text and Caption* ($\mathcal{S}_t$; $20.7\%$): These images lack visual information and solely consist of text heavily correlated with the caption. Many book covers constitute this type. These images would simply incentivize the model to learn the problem of optical character recognition.

The above classification is based on our manual judgement of the correlation between the various features and caption. In the next section, we use these findings to motivate and propose our data curation approach, `T-MARS`. Note that our experiments on text detection in § 5.3 further confirm that the estimated proportions based on our pilot study hold even at 64M scale LAION data subsets.

## 4 METHOD

Our pilot study in the § 3 revealed that a significant portion of the dataset consists of images for which text is the sole feature associated with the caption. Intuitively, these images encourage the model to solve optical character recognition in order to align the image and caption representations. Considering our downstream goal of learning better visual representations, it is natural to filter out such images. However, simply removing images that contain text matching the caption, is not be optimal due to the presence of images with both visual and text features as seen in our pilot study.

**Task**: Consider a pretraining image-caption dataset $\mathcal{S} = \{(i, t)\}^n$, used to train CLIP (Radford et al., 2021) style models using contrastive learning. Given a fixed computation budget (number of training iterations), our goal is to find a subset of the dataset $\hat{\mathcal{S}} \subset \mathcal{S}$, such that models trained on $\hat{\mathcal{S}}$ have higher zero-shot accuracy on downstream tasks (such as image classification) than those trained on $\mathcal{S}$.

**CLIP similarity score**: Given image-caption pair $(i, t)$, the CLIP score refers to the cosine similarity between the embeddings of the image and the caption, i.e., $f(i)^\top g(t)/\|f(i)\|_2\|g(t)\|_2$.

### 4.1 `T-MARS` : TEXT-MASKING AND RE-SCORING

Based on the above hypothesis, we propose a simple and scalable approach, `T-MARS` , which focuses on evaluating the similarity of *only the visual features* in an image with its corresponding caption. `T-MARS` first masks out the text in the image and then calculates the similarity score between the

masked image and the caption using a pre-trained CLIP model. By filtering out images with low masked similarity scores, we retain only those samples where visual features correlate with text.

1. *Text Detection*: We apply a text detection algorithm (FAST (Chen et al., 2021)) that identifies the bounding boxes of text regions in the image (Figure 1). Notably, text detection focuses on localizing text positions in the image rather than recognizing or reading the text itself. This key distinction allows our approach to be an order of magnitude more scalable compared to text recognition-based filtering methods (Radenovic et al., 2023).

2. *Text Masking*: Mask the text regions by replacing it with average color of the surrounding pixels.

3. *Re-Scoring and Filtering*: Using a pre-trained CLIP model, we calculate the cosine similarity between the masked image and the original caption. Finally, we simply filter out 50 percent of the datapoints that have the lowest similarity scores between the masked image and the caption. One can also choose to filter the datapoints based on a threshold score on the cosine similarity between the masked image and original caption.

We filter out $50\%$ of the pool and use a simple inpainting technique of neigbouring pixel colors to simplify design choices and they indeed serve us well. Note that we use the corresponding original images for training on the filtered subset, and not the masked images themselves. Algorithm Box 1 details T-MARS .

We first highlight the empirical effectiveness of T-MARS in Section 5.3. Later, in Section 6, we (1) show that T-MARS indeed works as intended, filtering out the images with only text features, while retaining those with both visual and text features, and (2) verify that all inputs with visual features have high positive utility and must be retained. On the other hand, images with only text features hurt as much as mislabeled ones.

---

**Algorithm 1** T-MARS

**Input:** Dataset $\mathcal{S} = \{i, t\}^n$, score function $\ell$, image masking function $m$
**Output:** Filtered Pool $\tilde{\mathcal{S}}$
// Step 1: Text-Masking
**for** $k = 0 \ldots n - 1$ **do**
$\quad \tilde{i}_k = m(i_k)$
**end for**
// Step 2: Re-Scoring
**for** $k = 0 \ldots n - 1$ **do**
$\quad s_k = \ell(\tilde{i}_k, t_k)$
**end for**
$\alpha = \text{Median}\left(\{s_k\}_{k=1}^n\right)$
return $\tilde{\mathcal{S}} = \{(i_k, t_k) \mid s_k \geq \alpha\}$

---

### 4.2 CONTRIBUTED BASELINES

Here we briefly describe two baseline filtering approaches, which we propose by drawing insights from hard example mining literature in the supervised training setup (a more detailed description can be found in Appendix A). As we show later in § 5.3, our proposed baselines themselves outperform the exisiting data curation approaches for training visual language models.

**C-SSFT** Maini et al. (2022) proposed the SSFT (Second-Split Forgetting Time) to identify mislabeled examples in a dataset by fine-tuning a converged model on validation data, and observing which examples change their predicted label the earliest. Given the absence of a converged model in webscale learning, we use a pre-trained model from OpenCLIP (Ilharco et al., 2021) and finetune for $n = 5$ epochs on the Conceptual-Captions dataset with a learning rate of $1e^{-5}$. We then calculate the average cosine similarity for all examples during the fine-tuning (forgetting) phase and rank examples based on the average similarity score, retaining only the highest-scoring ones.

**C-RHO** Mindermann et al. (2022) proposed RHO loss to prioritize the training on examples that are worth learning, but not yet learned. In classification, at every epoch, the method calculates the difference between the model's training loss on a given data point and a validation model's loss on it. Examples with low validation loss, but high training loss are prioritized. We propose C-RHO adapted for web-scale. (1) Rather than using the training loss, we utilize the model's image-caption similarity score; (2) We train our model for one epoch on the entire dataset to calculate the training loss and use a model trained on CC3M dataset as the validation model. Then, we calculate the difference in training and validation similarity scores to rank examples and only keep the top 50% of examples for training.

Table 1: Zero-shot accuracies for models trained on filtered subsets of the original LAION dataset when evaluated on a suite of 23 benchmark datasets (§ 5.2). Rows in 'orange' depict previous baselines (§ 4.3), those in 'white' depict our contributed baselines (§ 4.2), and those in 'green' depict our state-of-the-art method T-MARS (§ 4). ∩ denotes the intersection between two filtering strategies.

| Scale | Filtering | Dataset size | ResNet-50 | | | | ViT-B-32 | | | |
|---|---|---|---|---|---|---|---|---|---|---|
| | | | ImageNet | ImageNet dist. shifts | VTAB | Retrieval | ImageNet | ImageNet dist. shifts | VTAB | Retrieval |
| 16M | LAION | 100% | 16.63 | 15.04 | 24.20 | 16.79 | 09.39 | 08.46 | 19.83 | 12.58 |
| | CLIP Score (@ 50%) | 50.0% | 15.58 | 14.28 | 23.67 | 16.28 | 09.02 | 08.42 | 20.13 | 12.60 |
| | Text-Match | 86.4% | 17.83 | 15.83 | 24.63 | 17.11 | 10.16 | 08.89 | 20.63 | 12.84 |
| | C-SSFT | 90.0% | 17.49 | 15.61 | 24.90 | 17.31 | 10.10 | 08.94 | 19.67 | 13.26 |
| | C-RHO | 50.0% | 19.46 | 17.39 | 26.45 | 18.60 | 10.87 | 09.34 | 21.22 | 13.93 |
| | T-MARS | 50.0% | 20.25 | 17.71 | 26.50 | 18.45 | 12.09 | 10.35 | 22.64 | 14.15 |
| | T-MARS ∩ C-SSFT | 45.2% | 20.81 | 18.28 | 26.49 | 18.96 | 12.56 | 10.60 | 21.96 | 14.36 |
| | T-MARS ∩ C-RHO | 27.5% | 21.63 | 18.62 | 26.70 | 19.53 | 12.61 | 10.94 | 23.48 | 14.58 |
| 32M | LAION | 100% | 21.90 | 18.90 | 27.30 | 20.18 | 14.98 | 12.38 | 23.21 | 16.03 |
| | CLIP Score (@ 50%) | 50.0% | 20.84 | 18.79 | 25.71 | 19.54 | 14.69 | 12.86 | 22.81 | 15.32 |
| | Text-Match | 86.4% | 23.80 | 20.70 | 28.74 | 21.41 | 15.96 | 13.26 | 24.45 | 16.44 |
| | C-SSFT | 90.0% | 22.87 | 19.85 | 26.10 | 21.00 | 15.55 | 13.34 | 22.95 | 16.40 |
| | C-RHO | 50.0% | 25.44 | 21.81 | 27.65 | 22.61 | 16.76 | 13.98 | 25.60 | 17.48 |
| | T-MARS | 50.0% | 26.73 | 22.79 | 29.88 | 22.62 | 18.75 | 15.30 | 26.71 | 16.82 |
| | T-MARS ∩ C-SSFT | 45.2% | 26.89 | 22.83 | 28.81 | 22.99 | 19.18 | 15.86 | 27.13 | 17.82 |
| | T-MARS ∩ C-RHO | 27.5% | 27.20 | 23.30 | 30.30 | 22.77 | 19.15 | 15.86 | 26.93 | 18.04 |
| 64M | LAION | 100% | 26.34 | 23.24 | 29.09 | 23.91 | 20.37 | 17.97 | 27.85 | 18.83 |
| | CLIP Score (@ 50%) | 50.0% | 25.66 | 22.83 | 29.05 | 23.36 | 20.07 | 17.27 | 27.55 | 18.33 |
| | Text-Match | 86.4% | 29.11 | 24.94 | 30.35 | 25.75 | 23.11 | 19.04 | 28.82 | 19.37 |
| | C-SSFT | 90.0% | 28.15 | 24.13 | 29.73 | 25.58 | 21.80 | 18.20 | 27.69 | 19.54 |
| | C-RHO | 50.0% | 28.66 | 24.83 | 30.13 | 19.79 | 23.27 | 19.23 | 27.94 | 21.10 |
| | T-MARS | 50.0% | 32.47 | 27.52 | 33.05 | 24.99 | 25.78 | 21.05 | 31.69 | 20.52 |
| | T-MARS ∩ C-SSFT | 45.2% | 32.77 | 27.68 | 33.13 | 26.35 | 25.63 | 21.01 | 30.02 | 21.27 |
| | T-MARS ∩ C-RHO | 27.5% | 32.63 | 27.23 | 32.77 | 25.57 | 25.62 | 20.73 | 31.57 | 20.63 |

## 4.3 EXISTING BASELINES

**LAION filtering** The initial curation of the LAION-400M (Schuhmann et al., 2021) was based on filtering common-crawl samples with a CLIP similarity lower than 0.281 (using OpenAI's CLIP ViT-B/32). Samples with non-English captions are also filtered out.

**CLIP Score** We also investigate the use of stronger CLIP score thresholding by retaining image-caption pairs with high similarity to further reduce the size of the training pool by 50%. This would mean training multiple epochs on high CLIP-scored data, as opposed to a single epoch on all the data.

**Text Match** Radenovic et al. (2023) proposed removing all the images that contain text overlapping with the caption (5 continuous characters) to ensure that the model only focuses on visual features in the dataset. We skip the caption complexity and caption action filtering part, since it is shown to have a negative impact on accuracy in the original paper. Importantly, note that Text Match is $10\times$ more costly than just text masking, and the quality of text recognition in web images is so low that state-of-art recognition algorithms are unable to identify all text in the image correctly. On the other hand, text masking used in our work only requires detection, which is fast and accurate (Appendix D).

## 5 EXPERIMENTS

We evaluate various baselines (including those laid by this work) as well as our proposed approach T-MARS across 7 different data pools ranging from 2 million to 128 million. Our results showcase a linear scaling trend in the zero-shot accuracy gains over no data curation, highlighting the importance of incorporating data curation in practice as the data and compute are scaled.

## 5.1 DATA POOLS AND TRAINING CONFIGURATION

We first experiment on six different data pools ranging from 2M to 64M samples chosen from the LAION-400M dataset. Note that the compute budget (total training samples seen i.e. epochs ×

Table 2: Zero-shot accuracies for various filtering strategies on the `small` and `medium` pools of the DataComp benchmark. ∩ denotes the intersection between two filtering strategies. `T-MARS` outperforms the state-of-art on DataComp by a margin of 5% on the medium scale (ImageNet).

| Filtering | small (12.8M) | | | | | | medium (128M) | | | | | |
|---|---|---|---|---|---|---|---|---|---|---|---|---|
| | Dataset size | ImageNet | ImageNet dist. shifts | VTAB | Retrieval | Avg. | Dataset size | ImageNet | ImageNet dist. shifts | VTAB | Retrieval | Avg. |
| No filtering | 12.8M | 02.5 | 03.3 | 14.5 | 11.4 | 13.2 | 128M | 17.6 | 15.2 | 25.9 | 25.8 | 25.8 |
| Basic Filtering | 3.0M | 03.0 | 04.0 | 14.9 | 11.8 | 14.2 | 30M | 22.6 | 19.3 | 28.4 | 28.5 | 28.5 |
| LAION filtering | 1.3M | 03.1 | 04.0 | 13.6 | 09.2 | 13.3 | 13M | 23.0 | 19.8 | 30.7 | 29.2 | 29.2 |
| CLIP score (L/14 30%) | 3.8M | 05.1 | 05.5 | 19.0 | 11.9 | 16.4 | 38M | 27.3 | 23.0 | 33.8 | 32.8 | 32.8 |
| Text-Match | 3.5M | 05.7 | 06.2 | 18.9 | 12.0 | 17.3 | 34M | 29.4 | 24.7 | 34.4 | 26.0 | 34.3 |
| T-MARS | 2.5M | 06.4 | 06.7 | 20.1 | 13.4 | 17.9 | 25M | 33.0 | 27.0 | 36.3 | 29.4 | 36.1 |
| T-MARS ∩ C-RHO | 1.5M | 05.6 | 05.9 | 17.8 | 11.5 | 17.7 | 15M | 30.3 | 24.9 | 34.9 | 25.3 | 35.7 |
| T-MARS ∩ C-SSFT | 2.3M | 06.5 | 06.7 | 19.4 | 13.1 | 18.0 | 23M | 33.8 | 27.4 | 37.1 | 28.5 | 36.2 |

number of batches × batch size) is kept the same as the pool size. For example, for a 32M pool size, the total samples which can be seen during training is kept at 32M (i.e. 1 epoch over the whole dataset). In cases where filtering methods retain a smaller subset (say 16M samples) of the data pool, they get the advantage of running more iterations (2 epochs over 16M subset i.e. total 32M samples seen) over the chosen subset. Finally, we also experiment on the 12.8M (small scale) and 128M (medium scale) data pool of the recently released DataComp. We use the implementation of the Datacomp library to standardize the training process. We train both ResNet 50 and ViT-B-32 models with a batch size of 1024, using cosine learning rate with 200 steps of warmup at $5e^{-4}$. We use AdamW as the optimizer for training. All the experiments were performed on NVIDIA A6000 GPUs.

## 5.2 EVALUATION DATASETS

We extensively evaluate zero-shot accuracies on a suite of benchmarks considered in prior work (Radford et al., 2021; Wortsman et al., 2021): (a) ImageNet: a 1000-class image classification challenge (Russakovsky et al., 2015); (b) ImageNet-OOD: Six associated imagenet distribution shifts— ImageNetV2 (Recht et al., 2019), ImageNet-R (Hendrycks et al., 2020), ImageNet-A (Hendrycks et al., 2019), ImageNet-Sketch (Wang et al., 2019), ImageNet-O (Hendrycks et al., 2019), and ObjectNet (Barbu et al., 2019); (c) VTAB: 12 datasets from the Visual Task Adaptation Benchmark (Zhai et al., 2020), including Caltech101, CIFAR100, DTD, Flowers102, Pets, SVHN, Resisc45, EuroSAT, Patch Camelyon, Clevr Counts, Clevr Distance, KITTI and Sun397; and (d) Retrieval: 3 retrieval tasks of MSCOCO (Chen et al., 2015), Flickr (Young et al., 2014) and WinoGAViL (Bitton et al., 2022).

## 5.3 RESULTS

`T-MARS` gives impressive gains in accuracy over 23 downstream tasks of various types. Table 1 compares zeroshot accuracies of various data curation strategies under pool sizes of 16M, 32M and 64M. First, note that `T-MARS` consistently outperforms the baselines across the data pools. For example, on the 64M subset, `T-MARS` observes $6.4\%$ gains on ImageNet zeroshot accuracy over no filtering and $3.7\%$ gains over text matching. Similarly, `T-MARS` outperforms text-matching by $2.7\%$ in average accuracy over 6 ImageNet dist. shift datasets and by $2.78\%$ in accuracy over 13 vision tasks of the VTAB benchmark (Table 1). Results on 2M, 4M and 8M pool sizes are in Appendix E.

**Complementary data subsets** A very important observation from our work is that the data subsets filtered out by the three approaches proposed in our work have large fractions of exclusive subsets (see column data size). This observation translates into the fact that taking the intersection of data retained by different algorithms (`T-MARS` , `C-SSFT`, `C-RHO`) has additive benefits.

**Scaling Trends** An important consideration when proposing and evaluating any data filtering approach is whether or not the gains observed will continue to stay as the scale of data or compute grows. We present scaling trends for various techniques in Figure 3a, 1b which show that the gains in the zero-shot accuracy has a near linear slope as the data and compute are scaled exponentially (on the x-axis). This is extremely promising as it suggests that rather than gains saturating, gains offered by our method will grow logarithmically with the scale of the total data pool and compute.

**Higher accuracy using half the compute and half the data** We observe that selecting a better subset of data can be of higher utility compared to adding new unfiltered samples. For example,

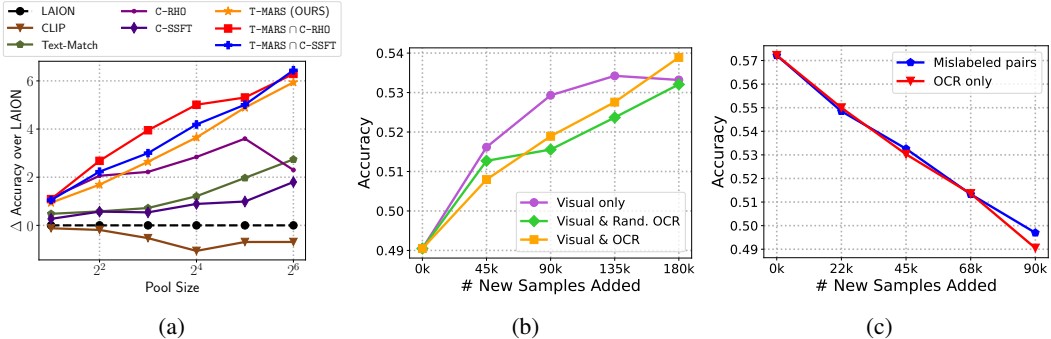

Figure 3: (a) Scaling curves depicting the increase in accuracy by training ResNet-50 models on filtered data versus training on the LAION dataset at different pool sizes. (b,c) We inspect the change in zero-shot accuracy on the Imagenette dataset when augmenting the training pool with new samples of various types. Images that contain visual features have similar utility, independent of the presence of text features; whereas those with only text features hurt as much as adding mislabeled examples.

`T-MARS ∩ C-RHO` filtered subset from the 32M pool gives an Imagenet accuracy of $27.20\%$ at a compute of 32M, which is around $1\%$ more than that of the LAION 64M data pool even at double the compute of 64M. This highlights the importance of incorporating data curation in practice, rather than expending additional compute on new unfiltered samples. In Section 6, we show a similar observation—there is higher utility in filtering out bad samples in comparison to adding new samples.

**State of Art on DataComp** We also evaluate `T-MARS` on the recently released data filtering benchmark DataComp (Table 2). For DataComp, we filter out the samples with a similarity score (between the masked image and original caption) below 0.281. Since the first release of `T-MARS`, there have been 10+ new methods on the Datacomp benchmark on the 'medium scale'. Notably, `T-MARS` and `T-MARS ∩ C-SSFT` are 2 of the top 3 entries on the leaderboard. Other top-performing entries require a mixture of multiple rules to achieve comparable or worse results than `T-MARS`. On the 'medium' scale, our proposed approach outperforms CLIP filtering by a large margin of $6.5\%$ and text-filtering by $4.4\%$. Datacomp leaderboard has another track 'bring your own dataset'. In this category, Nguyen et al. (2023) use the synthetic captions from BLIP (Li et al., 2023) to replace noisy web captions. Despite the expensive process of creating new data, rather than filtering existing subsets, `T-MARS` performs comparably on average, and even outperforms it by over 2% on Imagenet. This highlights the importance of filtering out the images without visual features. Based on the aforementioned scaling trends, our results portray an optimistic picture for practitioners with more compute budgets to implement `T-MARS` at the largest scale.

### 5.4 `T-MARS` EFFECTIVELY FILTERS TARGETED IMAGES

Recall the pilot study in Section 3. `T-MARS` filtered out a total of 250 images (out of 500 datapoints) and indeed works as expected by filtering out 95 of the 103 "text dominated" images, while also successfully retaining 46 out of 96 images that exhibit both visual and text features (in contrast, text match-based filtering retained only 21). CLIP score can be a noisy metric that is not well-calibrated across various images. Consequently, we also observe that in addition to removing text-dominated images, `T-MARS` also filtered out 76 of the 234 images with visual features only, because of their low alignment with the caption. That said, we do note that simply filtering based on CLIP score without masking (CLIP Score row, Table 1) performs even worse than no filtering, highlighting the significance of incorporating masking in `T-MARS`. We discuss further details in Appendix D.

## 6 WHAT TYPE OF DATA CONFERS GOOD VISUAL REPRESENTATION?

In this section, we utilize the characterization of data types in the LAION dataset from § 3 to simulate a similar data composition in a controlled experiment and evaluate the utility of each data type for improving visual features.

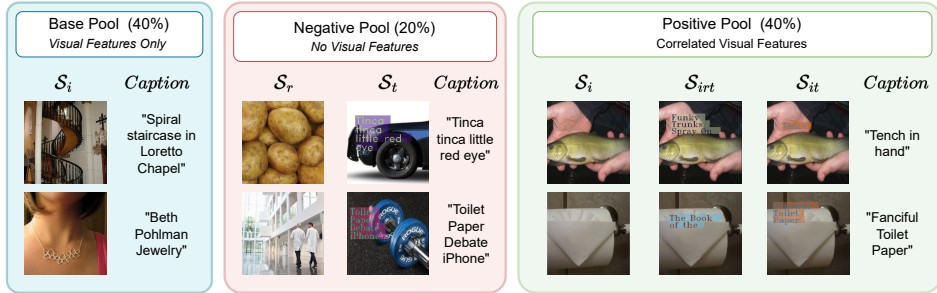

Figure 4: A representation of the various pools of data created for the synthetic experiments that evaluate example utility. **(a) Positive Pool**: Samples in category $\mathcal{S}_i$ are original image-caption pairs from the Imagenet-Captions dataset. Samples in $\mathcal{S}_{irt}$, are created by overlaying a random caption from the LAION dataset over the original image. Samples in $\mathcal{S}_{it}$ are created by overlaying the original caption onto the original image. **(b) Negative Pool**: To create a new sample for the category $\mathcal{S}_r$, we replace the image with a random image from the PACS dataset. Finally, samples in $\mathcal{S}_t$ are created by overlaying the original caption onto a random image from the PACS dataset.

**Experiment Protocol** We first create a synthetic dataset of image-caption pairs similar to the characterization of the LAION dataset in § 3. We detail the same in Figure 4 and Appendix C. Starting with a fixed base pool of 180k samples from the imagenet-captions dataset, we add new samples belonging to a particular data type and evaluate the accuracy of the trained model on the same. The number of training steps is the same across all training runs, and all results are averaged across 3 seeds. Evaluation is performed on the Imagenette dataset (a 10-class subset of Imagenet) (Howard).

**Results** In the local neighborhood of a fixed compute and data budget, we observe that different data types exhibit a linear relationship between the model's zero-shot accuracy and the number of samples from that type that are added. We define the utility of a data type at a given base configuration as $\mathcal{U}_{\text{type}} = {}^{\Delta\text{acc}}/_{\Delta\text{samples(in millions)}}$. Our main observations from Figure 3b and Figure 3c are:

1. Samples with only OCR feature ($\mathcal{U}_t = -0.89$) are as harmful as mislabeled ones ($\mathcal{U}_r = -0.8$).

2. If an image has useful visual features, then independent of the presence of useful ($\mathcal{U}_{it} = +0.27$), random ($\mathcal{U}_{irt} = +0.24$), or no OCR features ($\mathcal{U}_i = +0.23$), they have similar utility.

3. Removing bad examples has $3\times$ more utility than adding new good examples. This directly follows from the utility analysis of the OCR-only images, and those with visual features.

Overall, the above inferences further support the choices made in order to propose T-MARS. The utility of different data types confirms that we should retain samples that have both visual and text features in them, and naively removing all samples with text in them (such as in recent work by Radenovic et al. (2023)) is a sub-optimal strategy. Secondly, our results suggest that while scaling up data sizes has been a useful recipe for improving the quality of visual representation, the community should also focus on pruning off so-called 'bad examples' from the datasets, because pruning bad examples is significantly more useful than adding more examples.

## 7 LIMITATIONS AND CONCLUSION

We present a state-of-the-art data filtering approach, Text-Masking and Re-Scoring (T-MARS ), for web-scale datasets. T-MARS filters out examples where text dominates visual features in images, improving visual representation learning. Extensive evaluations on LAION and DataComp benchmark demonstrate that T-MARS outperforms competitive baselines by accuracy gains as high as $3.3\%$ and $6.5\%$ respectively. Data filtering can potentially introduce bias in the filtered subset. However, our text filtering approach is generic and does not incorporate any sort of group bias. Developing further nuanced metrics to rank the masked images beyond the CLIP score is an interesting direction for future work. While in this work, we create a *static* subset of the original corpus and perform multiple epoch training over the same, future work may benefit by assessing the utility of different data points in a dynamic fashion and refreshing the data pool with samples worth learning.

ACKNOWLEDGEMENTS

We thank the Datacomp and OpenCLIP teams for their code base that was used for training models in our work. ZL gratefully acknowledges the NSF (FAI 2040929 and IIS2211955), UPMC, Highmark Health, Abridge, Ford Research, Mozilla, the PwC Center, Amazon AI, JP Morgan Chase, the Block Center, the Center for Machine Learning and Health, and the CMU Software Engineering Institute (SEI) via Department of Defense contract FA8702-15-D-0002, for their generous support of ACMI Lab's research. AR gratefully acknowledges support from Open Philanthropy, Google, Apple and Schmidt AI2050 Early Career Fellowship. SG is supported by funding from the Bosch Center for Artificial Intelligence. PM is supported by funding from the DARPA GARD program.

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

# A    CONTRIBUTED BASELINES

In this work, apart from `T-MARS`, we also propose two competitive baselines `C-SSFT` and `C-RHO` which we described briefly in § 4.2. Here, we describe both the approaches in further detail.

## A.1    CONTRASTIVE SSFT (C-SSFT )

Maini et al. (2022) proposed the SSFT (Second-Split Forgetting Time) to identify mislabeled examples in a dataset by fine-tuning a (converged) model on a validation data, and observing which examples change their predicted label (to some other wrong label) the earliest. They show that mislabeled examples are forgotten the first i.e. change their predicted label to a wrong one during finetuning on some held-out validation set.

However, in the context of contrastive learning, two important differences arise—(1) there is no notion of convergence due to training for a limited number of epochs; (2) the concept of flipping of prediction is absent since contrastive loss only involves batch-specific loss.

To adapt the principle of forgetting to curation for web-datasets, in a scalable fashion, we propose Contrastive-SSFT (`C-SSFT`).

1. Given the absence of a converged model in webscale learning, we use a pre-trained model from OpenCLIP (Ilharco et al., 2021) and finetune for 5 epochs on some validation dataset.

2. For each training example in the pretraining dataset, we calculate it's average cosine similarity in the finetuning (forgetting phase) after every epoch. Finally, we rank the examples based on the average similarity score, retaining only the highest-scoring ones. Intuitively, examples with the low average cosine similarity score correspond to those which are forgotten (image and caption embedding misaligned) the earliest.

The following equation explains the procedure of estimating the forgetting scores for data points.

$$\texttt{C-SSFT} = \sum_{i=1}^{n} \mathcal{M}_{CC3}^{i}(i, t),$$

where $\texttt{C-SSFT}(i, t, \mathcal{S})$ is the score for an image-caption pair $(i, t)$ in a dataset $\mathcal{S}$, and $\mathcal{M}_{\mathcal{S}}^{n}$ is the similarity score based on a model trained for $n$ epochs on second-split validation dataset $\mathcal{S}$. $\mathcal{P}$ indicates the use of a pretrained model for initialization.

**Choice of validation dataset:** A critical question in `C-SSFT` is the choice of validation dataset, on which we finetune in the forgetting phase, as it directly influences the samples forgotten. Pretraining samples closer to the validation set will be inherently ranked higher by the design of our approach. For example, one can chose ImageNet as the validation set which will end up in `C-SSFT` ranking samples closer to the ImageNet distribution higher, consequently giving higher ImageNet accuracy (Table 3). However, this corresponds to *leaking the downstream evaluation set distribution into the pretraining corpus* (if not the exact evaluation samples), with which we disagree philosphically. Thus, in this work, for all the experiments on `C-SSFT`, we use a neutral Conceptual Captions (CC3M) as the validation set.

| | Second-split Validation Data | |
|---|---|---|
| LAION Pool Size | Conceptual Captions (CC-3M) | ImageNet |
| 4M | 18.5 | 19.45 |
| 8M | 22.32 | 25.26 |

Table 3: Choice of finetuning data for the forgetting phase is a critical question. Using ImageNet as the finetuning data can rank pretraining samples closer to ImageNet distribution higher and thus giving higher downstream accuracy. However, this corresponds to leaking evaluation set information in the pretraining corpus, even if not the exact samples. Thus, we use Conceptual Caption as the finetuning dataset in this work.

## A.2 CONTRASTIVE RHO (C-RHO )

Mindermann et al. (2022) proposed RHO loss to prioritize the training on examples that are worth learning, but not yet learned. In classification, at every epoch, the method calculates the difference between the model's training loss on a given data point and a validation model's loss on it. Examples with low validation loss, but high training loss are prioritized.

We propose `C-RHO` adapted for web-scale. (1) Rather than using the training loss, we utilize the model's image-caption similarity score; (2) We train our model for one epoch on the entire unfiltered pool and calculate the training loss (image-caption cosine similarity) in the end. (3) We use a model trained on Conceptual Captions (CC3M) dataset as the validation model.

Finally, we calculate the difference in training and validation similarity scores to rank examples and only keep the top 50% of examples for training.

$$\texttt{C-RHO}(i, t, \mathcal{S}) = \mathcal{M}_{CC3}^{32}(i, t) - \mathcal{M}_{\mathcal{S}}^{1}(i, t), \tag{1}$$

where $\texttt{C-RHO}(i, t, \mathcal{S})$ is the score for an image-caption pair $(i, t)$ in a dataset $\mathcal{S}$, and $\mathcal{M}_{\mathcal{S}}^{n}$ is the similarity score based on a model trained for $n$ epochs on dataset $\mathcal{S}$.

## B  ADDITIONAL RELATED WORK

**Visual Language Pre-training**    The field of vision-language pre-training has seen a surge of recent methods that attempt to pre-train CLIP-style models (Chen et al., 2022; Jia et al., 2021; Li et al., 2021; 2020; Pham et al., 2021; Wang et al., 2021).

**Neural Scaling Laws**    Recent works have shown a power law relation of test error with model size, compute, and training samples in the standard classification regime (Hestness et al., 2017; Zhai et al., 2022; Hernandez et al., 2021). Others have explored neural scaling laws for large language models and try to answer how the model size and training tokens should be scaled with compute (Hoffmann et al., 2022; Clark et al., 2022; Kaplan et al., 2020). Recently, Sorscher et al. (2023) explored the use of data curation in the classification settings to achieve accuracies beyond those predicted by these power laws. In this work, we consider data curation in the multimodal vision-language model training regime.

## C  SYNTHETIC EXPERIMENT SETUP DETAILS

For each example $j$, the dataset contains image ($x_j$), title ($y_j$), and metadata ($m_j$). We create image-caption pairs, $(i_j, t_j) = (x_j, \texttt{"Title:~~\{}y_j\texttt{\}~|~Metadata:~~\{}m_j\texttt{\}"}$ for each example. Then, depending on the category of data that a particular example should belong to, we modify the input in the following way:

1. $\mathcal{S}_r$: $(i_j, t_j)$ is replaced with $(\tilde{i}_j, t_j)$ by sampling $\tilde{i}$ from the PACS dataset.
2. $\mathcal{S}_i$: $(i_j, t_j)$ is used as it is.
3. $\mathcal{S}_{irt}$: $(i_j, t_j)$ is replaced with $(\tilde{i}_j, t_j)$ by overlaying the first four words of a random caption from the LAION dataset over $i_j$.
4. $\mathcal{S}_{it}$: $(i_j, t_j)$ is replaced with $(\tilde{i}_j, t_j)$ by overlaying the title corresponding to $t_j$ on $i_j$.
5. $\mathcal{S}_t$: $(i_j, t_j)$ is replaced with $(\tilde{i}_j, t_j)$ by overlaying the title corresponding to $t_j$ on a random image sampled from the PACS dataset.

**Creating Dataset Pool**    An important consideration for any controlled experiment is to make sure that the effect of adding different types of examples is compared by generating variants of the same example across different types. That is, when adding new samples from $\mathcal{S}_i, \mathcal{S}_{irt}, \mathcal{S}_{it}$, we want to ensure that they are variants of the same source example $(i_j, t_j)$. Similarly, we want to ensure the same when we remove examples from $\mathcal{S}_r, \mathcal{S}_t$ as well.

1. Base Data Pool: This is a sample of 40% of the Imagenet-Captions Dataset, and only contains samples belonging to $\mathcal{S}_i$. An illustration of the same is presented in panel 1 of Figure 4.

Table 4: Analysis of the fraction of image-caption pairs from different types that were retained by each filtering algorithm studied in our paper, assessed based on a 500-example pilot study. The first column presents the total number of images from each category in the subset of the LAION dataset.

| Image-Caption Category | Total Images | Text-Match | C-SSFT | C-RHO | T-MARS | T-MARS ∩ C-SSFT | T-MARS ∩ C-RHO |
|---|---|---|---|---|---|---|---|
| Random ($\mathcal{S}_r$) | 18 | 1.00 | 0.59 | 0.41 | 0.29 | 0.24 | 0.12 |
| Visual only ($\mathcal{S}_i$) | 234 | 0.99 | 0.89 | 0.60 | 0.67 | 0.60 | 0.37 |
| Visual, Random OCR ($\mathcal{S}_{irt}$) | 49 | 0.89 | 0.80 | 0.62 | 0.71 | 0.51 | 0.42 |
| Visual & OCR ($\mathcal{S}_{it}$) | 96 | 0.22 | 0.93 | 0.65 | 0.48 | 0.45 | 0.30 |
| OCR only ($\mathcal{S}_t$) | 103 | 0.08 | 0.91 | 0.46 | 0.07 | 0.07 | 0.04 |

2. Negative Pool: This is a sample of 20% of the Imagenet-Captions Dataset. For each example $(i_j, t_j)$ we create two copies of the sample for the types that do not contain any visual features—$\mathcal{S}_r, \mathcal{S}_t$. An illustration of the same is presented in panel 2 of Figure 4. The caption is preserved, but the images are either substituted with a random image (as in the case of $\mathcal{S}_r$), or with a random image with the title overlayed (as in the case of $\mathcal{S}_t$).

3. Positive Pool: This is a sample of the remaining 40% image-caption pairs from the Imagenet-Captions Dataset. For each example $(i_j, t_j)$ we create three copies of the sample for the types containing visual features—$\mathcal{S}_i, \mathcal{S}_{irt}, \mathcal{S}_{it}$. An illustration of the same is presented in panel 3 of Figure 4. The caption is preserved, but the images are either overlayed with a random text (as in the case of $\mathcal{S}_{irt}$), or with the caption title (as in the case of $\mathcal{S}_{it}$).

**Experiment Configuration** The base data pool is used for all experiments. For results in Figure 3b (experiments that evaluate the utility of images that contain visual features), we start with a base configuration of 180k samples from $\mathcal{S}_i$, and 90k samples from $\mathcal{S}_t$, and add varying sized subsets of new samples from $\mathcal{S}_i, \mathcal{S}_{irt}, \mathcal{S}_{it}$. For results in Figure 3c (experiments that evaluate the utility of random, and text-only images), we start with a base configuration of 180k samples from $\mathcal{S}_i$, and add varying-sized subsets of samples from $\mathcal{S}_r, \mathcal{S}_t$. Note that the final configuration for $\mathcal{S}_t$ is the same as the initial configuration for the graphs in Figure 3b. This is done in order to ensure that the model has the incentive to learn text features in order to perform well on the pre-training task. Only then can we evaluate if adding images with text features is hurtful to the model's learning or not? In the absence of text-only images, we find that the model is able to easily treat the text features as random features, defeating the purpose of the experiment. Finally, we add varying fractions of image-caption pairs from the positive pool to evaluate the utility of each data type. We train a randomly initialized ViT-B-32 vision encoder with a pre-trained RoBERTa text encoder for 120 steps of warmup followed by a cosine schedule with a maximum learning rate of $1e^{-3}$. The number of training steps is the same across all training runs (fixed at 600 steps at a batch size of 1024). Results for the same are presented in the main paper in Figure 3.

## D  DATA REMOVED BY VARIOUS CURATION APPROCHES

### D.1  HYPER-PARAMETER SEARCH FOR BASELINES

An important question that arises when using score-based filtering metrics like C-RHO , C-SSFT and T-MARS is how to select the score threshold above which a sample is retained, which consequently dictates the filtered subset size. In our experiments, for the 3 data filtering metrics above, we did a hyper-parameter search over filtered subset size in the grid $\{90\%, 75\%, 60\%, 50\%\}$ of the original unfiltered pool size. For C-SSFT , retaining $90\%$ samples i.e. removing only $10\%$ of the data worked the best, which is in line with the observations in Maini et al. (2022). For C-RHO and T-MARS , we observe that filtering out $50\%$ of the data worked the best. For Text-Matching, we used the criteria of Radenovic et al. (2023), where a sample is filtered out if 5 characters in OCR match with the caption.

### D.2  VALIDITY OF THE SAMPLE SIZE

Recall that our pilot study constituted only 500 samples. One may be concerned about the validity of the estimated proportions given the small sample size. We assess the error rate of the estimated proportions below.

1. If the sampling was done in a random and unbiased way, 500 samples is enough to estimate the proportions (with 95% confidence) to lie within a 5% margin of error. This can be understood as a problem of estimating the probability of an event occurring when a random variable is sampled from a binomial distribution. In particular, if $n$ is the required sample size for each category. $Z$ is the Z-score corresponding to the desired confidence level (for 95% confidence, $Z \sim 1.96$). $p$ is the estimated proportion of the category in the population, and E is the margin of error (set to 5%) then $n = \frac{Z^2 \cdot p \cdot (1-p)}{E^2}$ where $p$ is the estimated proportion. This would mean that the maximum value of $n$ would be less than 400 (when p = 0.5).

2. Second, is the sample itself representative of the population? To make the sampling unbiased, we take 500 random samples from the entire LAION dataset. However, this may still have a high variance. We performed an additional study on two more randomly sampled subsets (of 100 examples) of the LAION, and find that the estimated proportions lie within 2-3% of the population proportion.

3. Finally, and most convincingly, we have empirical evidence to support that the estimated proportions actually hold up for the entire dataset. For instance, the number of inputs found by text-match algorithm for the 500 samples we chose, and the entire LAION dataset are approximately of the same proportion (see Table 4).

### D.3    VISUALIZING THE DATA REMOVED

Recall in Section 5.4, we discussed the number of samples removed by various filtering metrics from each category in our 500 image pilot study. Table 4 lists the fraction of samples from each category retained by various metrics. Figure 5 shows the images that were filtered out by text matching metric but retained by T-MARS . Recall that these would correspond to the cases where both the visual and text features are correlated with the caption. Figure 6 shows the images that were filtered out by both metrics. Finally, in Figure 7, Figure 8, Figure 9 and Figure 10 we share some of the samples removed by C-SSFT and C-RHO metrics.

### D.4    FAILURE CASES FOR TEXT-MATCHING

Recall that text-matching requires recognizing the text, which is up to an order of magnitude more computationally expensive than text-detection. However, additionally, we observe that text recognition has a lot of failure cases as well, where although the text is detected correctly, the recognition process fails to read correctly, as shown in Figure 11. In line with the previous works Radenovic et al. (2023), we used the MMOCR library for text recognition. These failure modes may be addressed with the use of much more expensive transformer-based text-recognition models like Text Spotting Transformers[1], albeit at a prohibitively high cost for webscales considered in this work.

## E    ADDITIONAL RESULTS ON LAION

Recall that in Section 5.3, we observed a linear scaling trend of zero-shot accuracies as we varied the pool size (LAION subset) from 2M to 64M. In Table 5, we share the detailed zero-shot accuracies for models trained on various subsets.

---

[1] https://github.com/mlpc-ucsd/TESTR

| Image | Caption | Image | Caption |
|---|---|---|---|
|  | One Missed Call (Combo HD DVD and Standard DVD) [HD DVD] |  | D'Andrea Snarling Dog Brain Nylon Guitar Picks 72 Pack Refill (Purple, 0.60mm) |
|  | adidas Sacramento Kings Jimmer Fredette Infant Replica Road Jersey |  | 12V USB Car Air Humidifier for Car and Computer pictures & photos |
|  | Hostgator Web Hosting |  | The Girl with the Dragon Tattoo - Stieg Larsson |
|  | Budapest PopOut Map: pop-up city street map of Budapest city center - folded pocket size travel map. .. |  | Henry's Hard Soda - Hard Orange |
|  | Shower Gel - Vanilla Bean Noel |  | Factory Price For Tube Sealing Machinery - Tube Fill And Seal Machine – Maxwell |

Figure 5: Samples removed by Text-Matching but retained by our proposed approach `T-MARS` . These correspond to the images where both the visual and text features are correlated with the caption .

| Image | Caption | Image | Caption |
|-------|---------|-------|---------|
| | setup your professio nal ★wordpress★ site within ►6hours◄ | | Simple Thai Basil – Coconut Chicken |
| | IAM Overview and Se lf-assessment Exerci se | | Minnesota Department of Transportation |
| | Curious Critters Clu b - Logo English Kid s Longsleeve T-Shirt by kilopop's Artist Shop | | "VRP A215 1/8 ""Game changer"" Piston (2) (1.3mm x 6 Hole) (H igh Pack)" |
| | Le second livre de l a jungle - Kipling | | WEB ART DESIGN GOURM ANDISE CHOCOLAT TENT ATION PLAISIR 100 |
| | Entranement intensif aux tests d'ap titude IFSI : Nombre s, Lettres, Formes, Dominos, Cartes | | Quilting Internation al Magazine, 1990 - QM |

Figure 6: Samples removed by both text-matching and `T-MARS` . Here only the text features are correlated with the caption
.

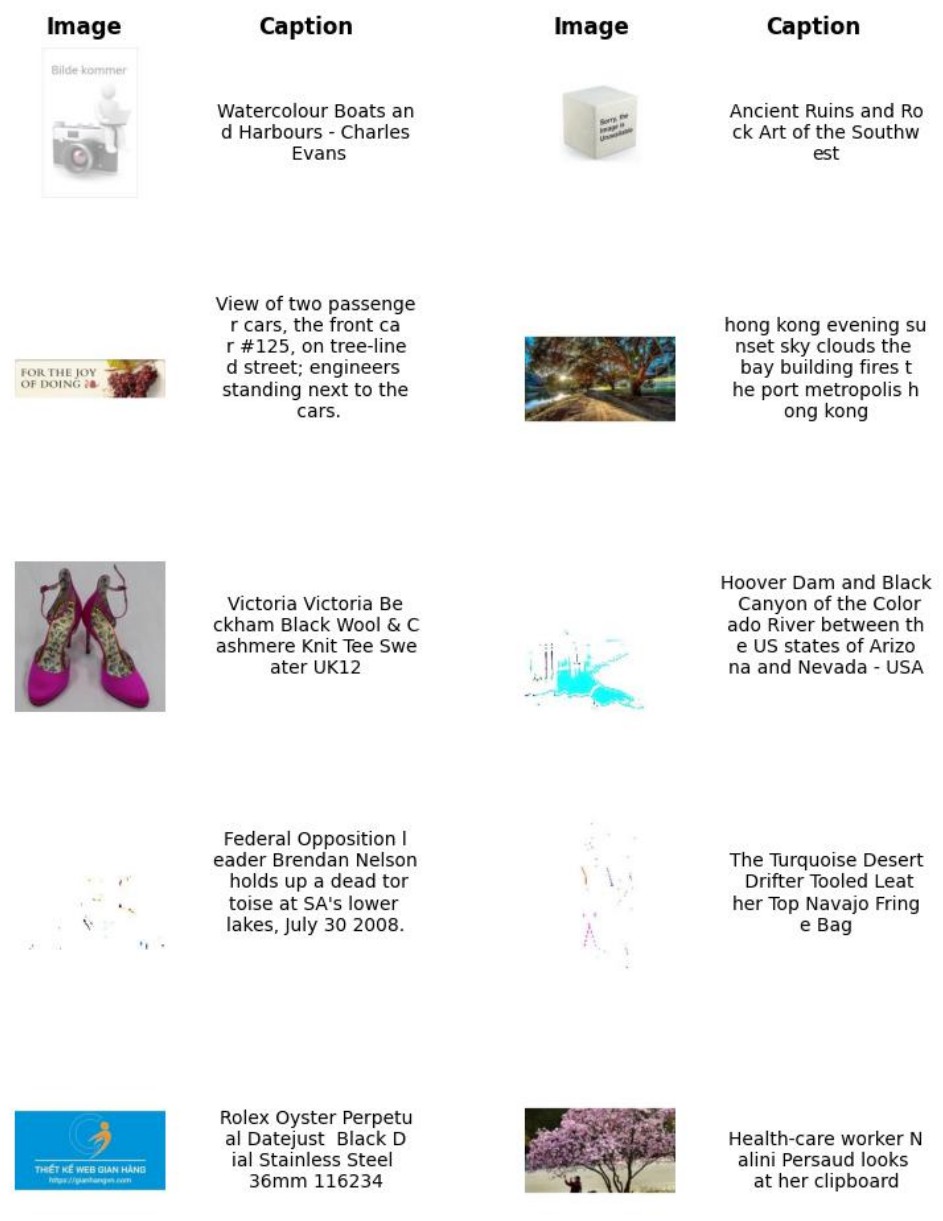

Figure 7: Samples with the lowest score (and filtered out) based on the `C-SSFT` metric.

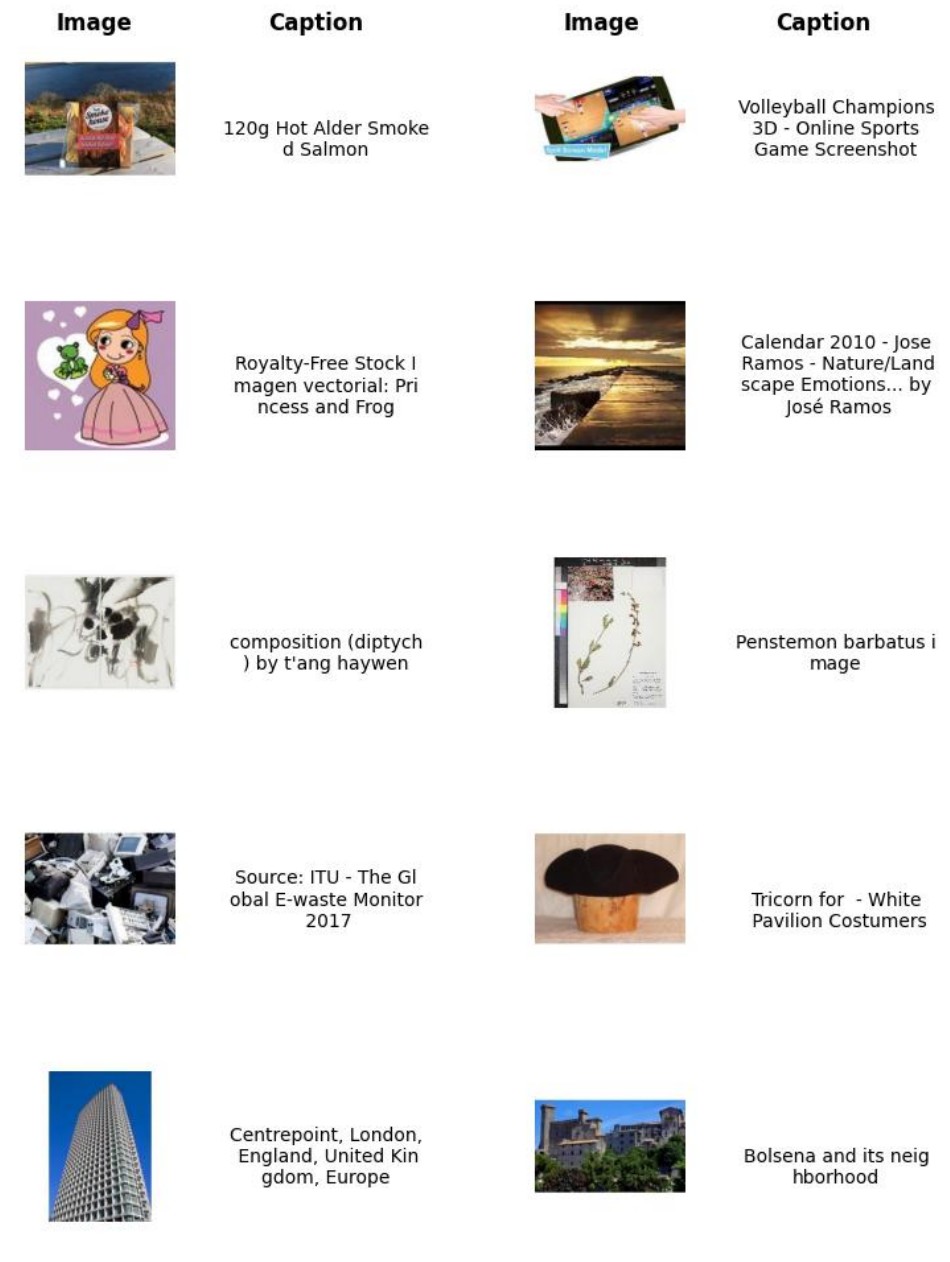

Figure 8: Additional (random) samples filtered out by the `C-SSFT` metric
.

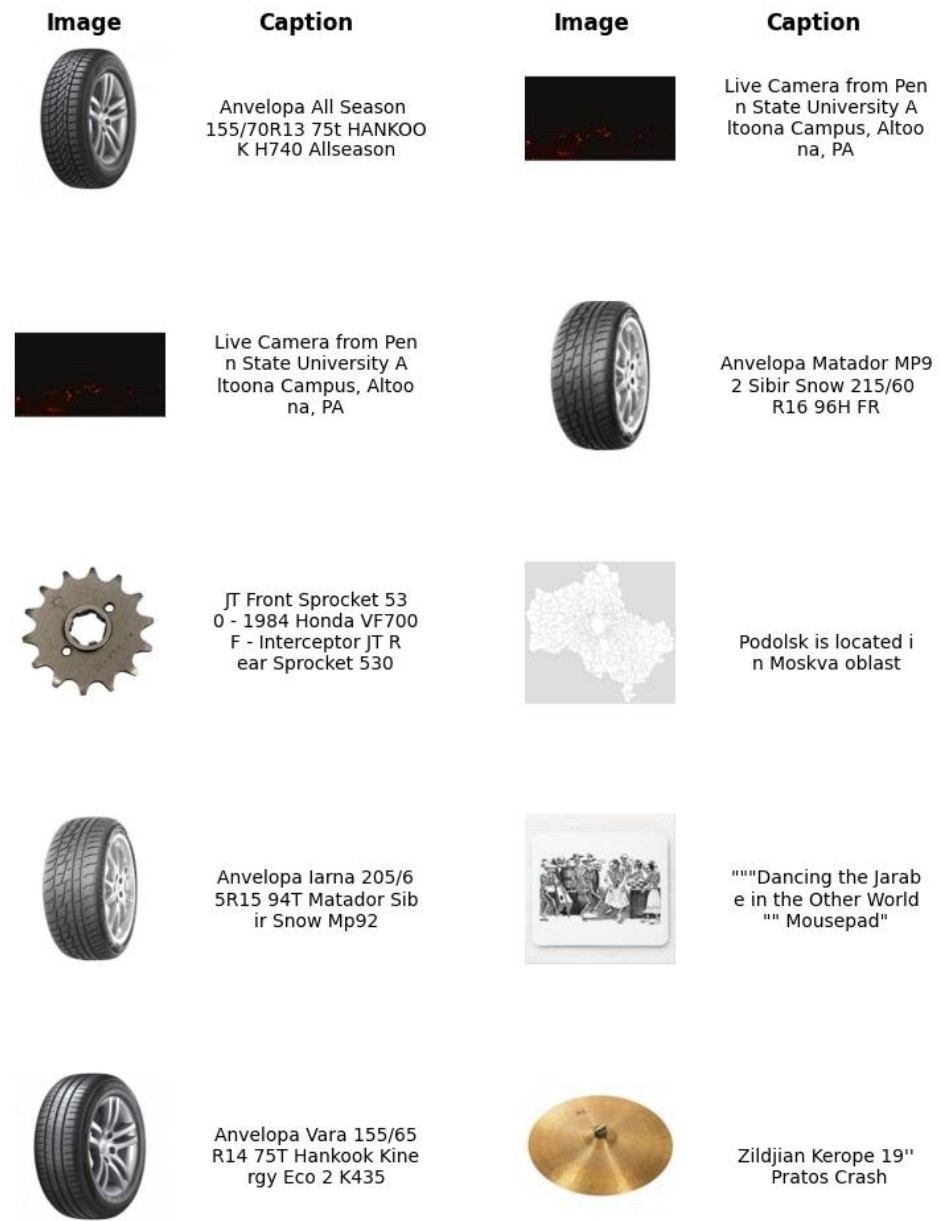

Figure 9: Samples with the lowest score (and filtered out) based on the `C-RHO` metric
.

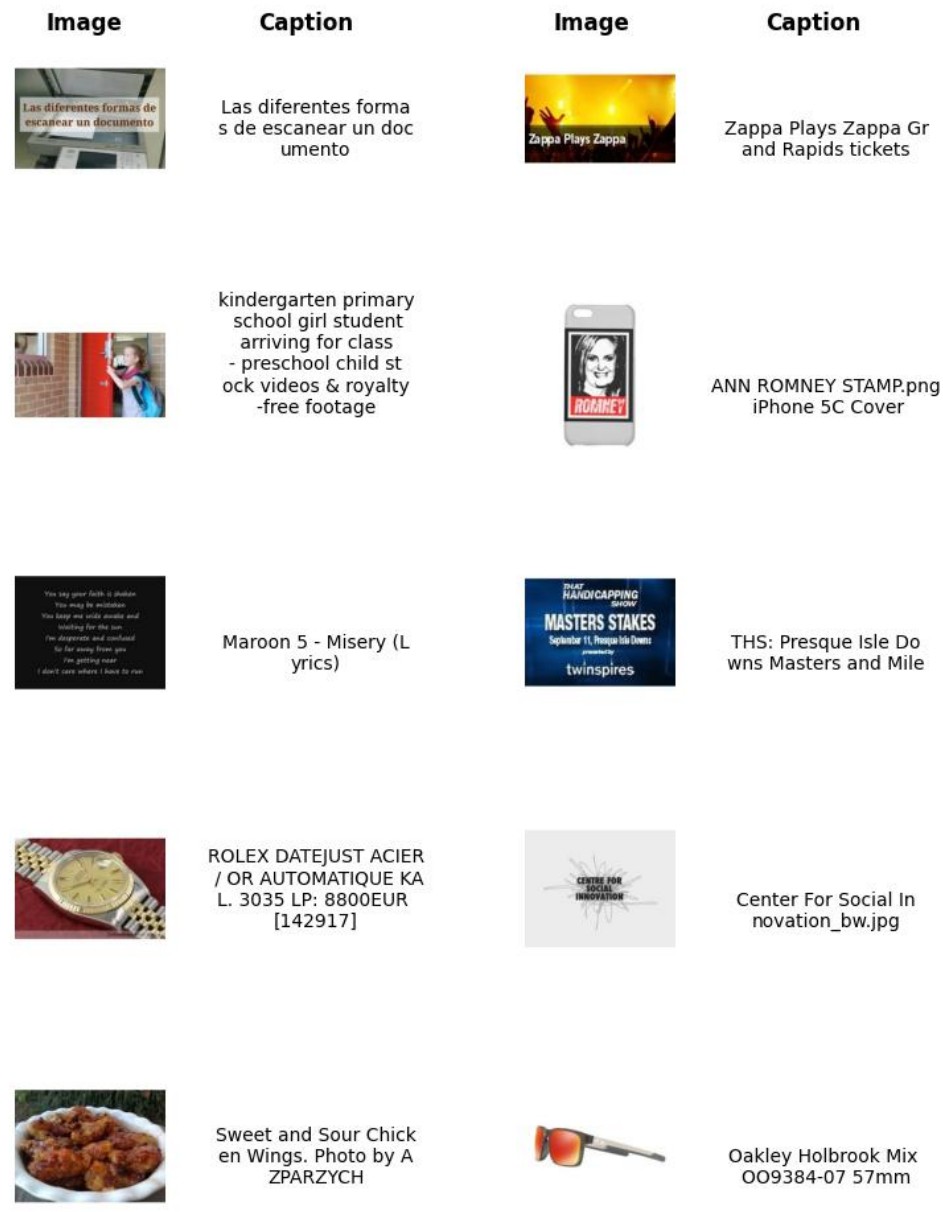

Figure 10: Additional (random) samples filtered out by the `C-RHO` metric
.

| Image | Caption | Image | Caption |
|-------|---------|-------|---------|
|  | WINSTEAD, JOSEPH R - Boone County, Arkansas \| JOSEPH R WINSTEAD - Arkansas Gravestone Photos |  | naseberry treatment cream |
|  | HARMONY DAY - THEME BOOK! - All Ages |  | Magical Mystery Tour (1967) (Movie) |
|  | Vicki Brewer Brotherhood of Light Egyptian Tarot [With Booklet] |  | Filson Ballistic Nylon Travel pack Navy |
|  | Falcon Publishing Mountain Biking Eastern Ny |  | The Spanish Telefonica logo |
|  | Thanksgiving Placemat Crafts and Napkin Ring Crafts for Kids |  | Streets Of Your Town Sheet Music |

Figure 11: Some failure cases of text-matching, due to failure of the text recognition process. All these images have text overlapping with the caption and should have been filtered out ideally, but end up being retained as the text recognition fails to read the text correctly.

.

Table 5: Zero-shot accuracies for models trained on filtered subsets of the original LAION dataset when evaluated on a suite of 17 benchmark datasets (§ 5.2). Rows in 'orange' depict previous baselines (§ 4.3), those in 'white' depict our contributed baselines (§ 4.2), and those in 'green' depict our state-of-the-art method T-MARS (§ 4). ∩ denotes the intersection between two filtering strategies.

| | | | ResNet-50 | | | | ViT-B-32 | | | |
|---|---|---|---|---|---|---|---|---|---|---|
| Scale | Filtering | Dataset size | ImageNet | ImageNet dist. shifts | VTAB | Retrieval | ImageNet | ImageNet dist. shifts | VTAB | Retrieval |
| 2M | LAION | 100% | 02.57 | 03.50 | 14.82 | 09.27 | 01.21 | 02.04 | 13.42 | 08.23 |
| | CLIP Score (@ 50%) | 50.0% | 02.46 | 03.62 | 14.61 | 10.09 | 01.26 | 02.40 | 13.75 | 07.92 |
| | Text-Match | 86.4% | 03.05 | 03.78 | 15.97 | 09.28 | 01.35 | 02.45 | 13.05 | 08.90 |
| | C-SSFT | 90.0% | 02.85 | 03.65 | 15.41 | 09.64 | 01.38 | 02.38 | **14.96** | 08.76 |
| | C-RHO | 50.0% | **03.70** | 04.41 | 15.67 | 10.84 | 01.46 | 02.54 | 14.85 | 09.25 |
| | T-MARS | 50.0% | 03.51 | 04.18 | 14.86 | **10.87** | 01.40 | 02.41 | 12.98 | **09.77** |
| | T-MARS ∩ C-SSFT | 45.2% | 03.62 | 04.48 | 16.59 | 09.98 | 01.60 | 02.61 | 14.72 | 09.59 |
| | T-MARS ∩ C-RHO | 27.5% | **03.70** | **04.58** | **16.80** | 10.20 | **01.72** | **02.77** | 14.79 | 09.63 |
| 4M | LAION | 100% | 07.06 | 07.06 | 17.79 | 11.72 | 03.05 | 03.79 | 16.18 | 10.00 |
| | CLIP Score (@ 50%) | 50.0% | 06.86 | 07.40 | 18.07 | 11.95 | 03.20 | 04.00 | 15.71 | 09.36 |
| | Text-Match | 86.4% | 07.66 | 07.39 | 18.42 | 12.28 | 03.51 | 04.30 | 16.70 | 09.60 |
| | C-SSFT | 90.0% | 07.64 | 07.44 | 18.94 | 12.22 | 03.42 | 04.19 | 16.66 | 09.73 |
| | C-RHO | 50.0% | 09.12 | 08.67 | 20.73 | 13.73 | 03.60 | 04.34 | 16.38 | 10.63 |
| | T-MARS | 50.0% | 08.77 | 08.69 | 20.94 | 13.67 | 04.04 | 04.64 | **17.10** | 11.59 |
| | T-MARS ∩ C-SSFT | 45.2% | 09.30 | 08.80 | 19.12 | 13.45 | 04.22 | 04.63 | 17.04 | 11.27 |
| | T-MARS ∩ C-RHO | 27.5% | **09.75** | **09.20** | **21.41** | **14.17** | **04.28** | **05.05** | 16.20 | **11.69** |
| 8M | LAION | 100% | 11.62 | 10.77 | 21.74 | 14.02 | 05.57 | 05.81 | 17.32 | 10.95 |
| | CLIP Score (@ 50%) | 50.0% | 11.13 | 10.60 | 21.87 | 13.88 | 05.80 | 05.92 | 17.45 | 10.90 |
| | Text-Match | 86.4% | 12.38 | 11.35 | 22.64 | 14.03 | 06.16 | 06.19 | 17.89 | 10.94 |
| | C-SSFT | 90.0% | 12.19 | 11.36 | 20.88 | 14.54 | 05.92 | 06.22 | 17.60 | 11.21 |
| | C-RHO | 50.0% | 13.86 | 12.94 | 22.12 | 15.90 | 06.55 | 06.43 | 18.55 | 11.77 |
| | T-MARS | 50.0% | 14.10 | 12.97 | 22.20 | 16.05 | 07.20 | 07.28 | 19.02 | 12.83 |
| | T-MARS ∩ C-SSFT | 45.2% | 14.65 | 13.01 | 22.35 | 16.10 | 07.54 | 07.22 | 19.18 | 12.66 |
| | T-MARS ∩ C-RHO | 27.5% | **15.60** | **13.09** | **22.85** | **16.33** | **07.65** | **07.39** | 18.63 | **13.17** |
| 16M | LAION | 100% | 16.63 | 15.04 | 24.20 | 16.79 | 09.39 | 08.46 | 19.83 | 12.58 |
| | CLIP Score (@ 50%) | 50.0% | 15.58 | 14.28 | 23.67 | 16.28 | 09.02 | 08.42 | 20.13 | 12.60 |
| | Text-Match | 86.4% | 17.83 | 15.83 | 24.63 | 17.11 | 10.16 | 08.89 | 20.63 | 12.84 |
| | C-SSFT | 90.0% | 17.49 | 15.61 | 24.90 | 17.31 | 10.10 | 08.94 | 19.67 | 13.26 |
| | C-RHO | 50.0% | 19.46 | 17.39 | 26.45 | 18.60 | 10.87 | 09.34 | 21.22 | 13.93 |
| | T-MARS | 50.0% | 20.25 | 17.71 | 26.50 | 18.45 | 12.09 | 10.35 | 22.64 | 14.15 |
| | T-MARS ∩ C-SSFT | 45.2% | 20.81 | 18.28 | 26.49 | 18.96 | 12.56 | 10.60 | 21.96 | 14.36 |
| | T-MARS ∩ C-RHO | 27.5% | **21.63** | **18.62** | **26.70** | **19.53** | **12.61** | **10.94** | **23.48** | **14.58** |
| 32M | LAION | 100% | 21.90 | 18.90 | 27.30 | 20.18 | 14.98 | 12.38 | 23.21 | 16.03 |
| | CLIP Score (@ 50%) | 50.0% | 20.84 | 18.79 | 25.71 | 19.54 | 14.69 | 12.86 | 22.81 | 15.32 |
| | Text-Match | 86.4% | 23.80 | 20.70 | 28.74 | 21.41 | 15.96 | 13.26 | 24.45 | 16.44 |
| | C-SSFT | 90.0% | 22.87 | 19.85 | 26.10 | 21.00 | 15.55 | 13.34 | 22.95 | 16.40 |
| | C-RHO | 50.0% | 25.44 | 21.81 | 27.65 | 22.61 | 16.76 | 13.98 | 25.60 | 17.48 |
| | T-MARS | 50.0% | 26.73 | 22.79 | 29.88 | 22.62 | 18.75 | 15.30 | 26.71 | 16.82 |
| | T-MARS ∩ C-SSFT | 45.2% | 26.89 | 22.83 | 28.81 | **22.99** | **19.18** | **15.86** | **27.13** | 17.82 |
| | T-MARS ∩ C-RHO | 27.5% | **27.20** | **23.30** | **30.30** | 22.77 | 19.15 | 15.86 | 26.93 | **18.04** |
| 64M | LAION | 100% | 26.34 | 23.24 | 29.09 | 23.91 | 20.37 | 17.97 | 27.85 | 18.83 |
| | CLIP Score (@ 50%) | 50.0% | 25.66 | 22.83 | 29.05 | 23.36 | 20.07 | 17.27 | 27.55 | 18.33 |
| | Text-Match | 86.4% | 29.11 | 24.94 | 30.35 | 25.75 | 23.11 | 19.04 | 28.82 | 19.37 |
| | C-SSFT | 90.0% | 28.15 | 24.13 | 29.73 | 25.58 | 21.80 | 18.20 | 27.69 | 19.54 |
| | C-RHO | 50.0% | 28.66 | 24.83 | 30.13 | 19.79 | 23.27 | 19.23 | 27.94 | 21.10 |
| | T-MARS | 50.0% | 32.47 | 27.52 | 33.05 | 24.99 | **25.78** | **21.05** | **31.69** | 20.52 |
| | T-MARS ∩ C-SSFT | 45.2% | **32.77** | **27.68** | **33.13** | **26.35** | 25.63 | 21.01 | 30.02 | **21.27** |
| | T-MARS ∩ C-RHO | 27.5% | 32.63 | 27.23 | 32.77 | 25.57 | 25.62 | 20.73 | 31.57 | 20.63 |

Original Image      Masked Image

Figure 12: Comparing the original and the masked image. Text detection and masking in general doesn't seem to cause major aberrations in the patches that have useful visual features.
.

## F    SOME EXAMPLES OF ORIGINAL V/S MASKED IMAGE

In Figure 12, we give some examples of original and masked image. The text-detection algorithm in general gives tight bounding boxes around the area with text. This ensures that image patches with useful visual features do not suffer a major aberration. However, developing better ways of masking the text is an interesting (orthogonal) direction of future work.

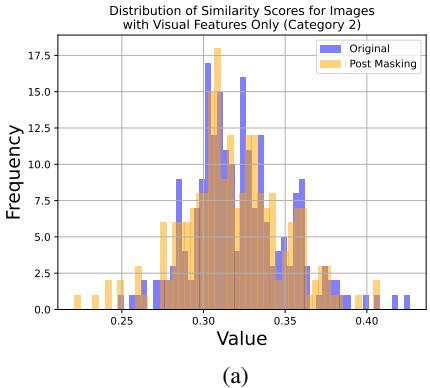 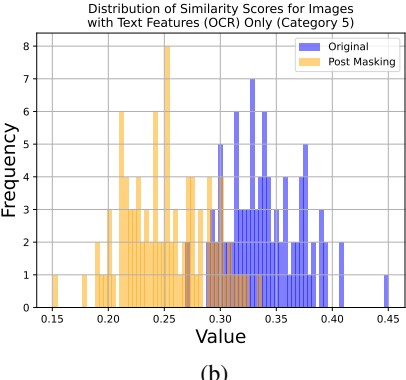

(a)                                                          (b)

Figure 13: Change in the distribution of CLIP similarity scores, before and after masking the text (OCR) over the images. (a) Distribution of scores of images with visual features only (Category 2, § 3); (b) Distribution of scores of images with text features only (Category 5, § 3)

## G  DISTRIBUTION OF SIMILARITY SCORES AFTER MASKING

`T-MARS` calculates the CLIP similarity score of data points after masking the text (OCR) over the images, and filters out image-caption pairs that no longer have high similarity. The goal of this masked-evaluation is to remove images that have only text features in them. But does the proposed masking strategy actually lower the CLIP score of the examples as intended? Recall our 500 example pilot study where we hand-labeled examples into various categories, like those with only visual features or those with only text features (§ 3). We calculated the CLIP similarity score of these image-caption pairs, before and after masking the text in the respective images. Figure 13a and Figure 13b show the histogram for the distribution of scores, before and after masking for images with visual features only (Category 2, § 3) and images with text features only (Category 5, § 3).

Our observations indicate that after masking, the CLIP score of images with just text features (OCR) drops significantly (indicating the will be filtered out by T-MARS). At the same time, CLIP scores of images with only visual features stays nearly the same. This demonstrates the efficacy of the filtering method for removing samples with only text features.

## H  DISCUSSION: CIRCUMVENTING TEXT FEATURE LEARNING

In this work, we consider a straightforward strategy of circumventing text feature learning by removing samples where the image-caption alignment was dominated by the text feature in the image (`T-MARS`). Identifying other ways to circumvent text features remains an open problem for future work, and we briefly discuss some other strategies in this section.

**Removing all inputs with text features**    Previous work by Radenovic et al. (2023) considers the simple idea of removing any image-caption pair where the image contains any text features that are correlated with the caption. This is a natural way to avoid learning of text features (OCR), however, removing all such image-caption pairs is suboptimal to model training as it also removes samples from the category that contains both image and text features as discussed in Section 5.3 under the TextMatch algorithm.

**Re-captioning Web-datasets**    Another alternate way to be able to utilize the knowledge contained in image-caption datasets, yet not learn text features is to re-caption the input text such that it does not utilize the OCR in the text in the image. This can be operationalized in a variety of ways—(i) identifying text OCR and modifying the input text to not contain text overlap with the identified text by using a paraphrase, (ii) using an off-the-shelf image-captioning model to provide captions for inputs on the web. The latter approach was, in fact, used in follow-up work of Nguyen et al. (2023) where the authors use a BLIP Li et al. (2023) model to provide synthetic captions for the web data, and subsequently pre-train their model on a mixture of synthetic and real captions. One reason why this method

works is that the original BLIP model was trained on datasets such as MS-COCO and NoCaps where the input images typically do not contain OCR text features. When analyzing the captions generated by the BLIP model as provided by the authors at `https://huggingface.co/datasets/thaottn/DataComp_medium_pool_BLIP2_captions`, we found that the BLIP model generated captions do not contain the OCR text in the images. This can be a useful strategy for circumventing text feature learning, however, as discussed in Section 5.3 we observe `T-MARS` performs 2% better than this approach on zero-shot evaluation on the Imagenet Dataset.

**Inpainting text features in image**    Just like modifying the text component of an image-caption pair, we may also alternately modify the input images to not contain the OCR. For the purposes of our work (`T-MARS` ), we only wanted to calculate the CLIP score post-masking and hence used a basic in-painting algorithm of in-filling the text region with the average score of the neighboring pixels. However, rather than throwing away the input samples, we may use generative models to in-paint the OCR regions and train on these image-caption pairs. This remains an avenue for future work to explore. However, we also have a few reservations about this approach. In particular, we chose not to train on masked images to avoid distribution shifts and loss in performance potentially due to the aberrations introduced by masking. Moreover, there is no reason to train on masked images from Category 5 (samples with only text features) because they will be data points where the text and the (now inpainted) input are uncorrelated. As far as Category 3 is concerned (samples with random text features, but correlated vision features), the uncorrelated texts are already ignored by the model (as also seen in Section 6). Finally, let us come to the most important category in question (Category 4 with both correlated text and visual features). Currently, we retain the unmasked input in such categories (that have both text and visual features). One proposal would be to only retain the visual features in these. On the flip side, keeping the text features can be helpful for retaining an understanding of certain visual features as well, such as brands and trademark logos (such as the Superman logo) which can be in-part textual, but also have a distinct visual pattern. An infilling algorithm will mask these patterns and potentially hinder the model's knowledge.

Overall, there remain various exciting avenues for alternatively circumventing text feature learning in vision-language models. Our work on `T-MARS` proposes one strong baseline for the same, and we hope that future work can build on our work and the insights provided in this discussion.

