# OpenReview forum: "T-MARS: Improving Visual Representations by Circumventing Text Feature Learning"
_ICLR.cc/2024/Conference — ICLR 2024 poster_

### Official Review · Reviewer_NY9s · 2023-10-14

**Soundness:** 4 excellent
**Presentation:** 4 excellent
**Contribution:** 4 excellent
**Rating:** 8
**Confidence:** 4

**Summary:**

This paper introduces T-MARS (Text Masking and Re-Scoring), a data-filtering method for curating image-text datasets. The method is built on the observation that a large portion of web-crawled image-text datasets such as LAION contain text that overlap significantly with the image (and often lack visual representations of what the text refers to). Intuitively, such datapoints could encourage models to prioritize learning OCR over learning the visual contents of the images. Their method, T-MARS attempt to filter such samples. Their experiments show strong empirical results, improving over strong baselines and competing methods in DataComp by a large margin. Overall, their method is simple, scalable and effective.

**Strengths:**

There are many strengths to this paper.

1. Firstly, understanding how to better design datasets is an important and timely problem, and many open problems remain. This paper presents a significant step forward in that direction. As such, I believe this paper would be of interest to many in the community and will have substantial impact for practitioners interested in building better multimodal models.
2. The proposed method is simple and novel, and can be easily applied to any data curation pipeline.
3. As shown by the authors, the method also scales well.
4. The experimental results are very strong, providing large gains in accuracy over strong baselines and existing data curation methods.
5. The paper is very clear and well written.

**Weaknesses:**

The main weakness I see in this paper is the lack of large scale experiments. However, I do not think this should count against the authors, for a few reasons. Firstly, running large-scale CLIP pre-training experiments can be prohibitively expensive for many institutions. Secondly, the authors present clear scaling trends that show that their approach holds great promise for larger scales.

**Questions:**

1. In Section 4.1, why exactly 50% of the pool is filtered?
2. I'm sometimes a bit confused by the choice of downstream evaluation tasks. In particular, the DataComp defines a clear set of 38 downstream tasks, yet the authors evaluate on only subsets of these tasks (and often not even the same subsets, e.g. table 1 is on 17 datasets, Table 2 doesn't show the average over the 38, and in Section 5.3 it says 23 datasets are used). Why the inconsistencies?
3. Why are some of the stronger baselines (including ones from the DataComp paper) not present in some tables?

---

> ### Author Response · Authors · 2023-11-20
>
> We thank you for your time, and valuable feedback, acknowledging “very strong results” and “promising scaling trends”. We are glad to see your recognition of our pilot study as a “significant step towards understanding multimodal datasets”.  Below we provide clarifications to the questions raised.
>
> ### **Why is exactly 50% of the data filtered**
> While designing the approach and experimenting on the LAION-400M dataset, we tried three choices for the amount of data to be filtered— 40%, 50%, and 75%, with 50% working out to be the best. Note that these experiments further filtered an already filtered LAION dataset. One can definitely vary this and use other filtering metrics, say based on a threshold CLIP score after text masking. For example, on DataComp (filtering on the common crawl),  we use a CLIP-similarity threshold of 0.281, which is similar to that used in prior work, such as LAION filtering. Fine-tuning the threshold may bring further gains, but running such parameter searches is beyond our compute scope. We have clarified this in the updated manuscript as well.
>
> ### **Downstream Datasets**
> We apologize for the oversight on our part and have updated Table 2 (Datacomp results) with an evaluation on all 38 downstream tasks. We have fixed the typo on the number of datasets in Table 1 as well. Our experiments in Table 1 were on quite a small scale initially, due to which we did not evaluate on some tasks like WILDS in Datacomp on which all models get random performance.
>
> ### **Some missing baseline from DataComp paper**
> The DataComp paper also considered an ‘image-based’ filtering approach that uses information from the Imagenet dataset (ImageNet $\cap$ CLIP Score). T-MARS outperforms this baseline as well, but we do not list it because we wanted to only consider methods that do not use any information of the downstream evaluation dataset to provide for a fair evaluation between all baselines.
>
> ---
>
> We again thank you for your time and hope that we have addressed the concerns. We are happy to provide any further clarifications if requested.

---

> > ### Comment · Reviewer_NY9s · 2023-11-21
> >
> > Thank you for addressing my comments. I have read the other reviews and the responses from the authors and stick to my original score.

---

### Official Review · Reviewer_3vr7 · 2023-10-31

**Soundness:** 3 good
**Presentation:** 3 good
**Contribution:** 2 fair
**Rating:** 6
**Confidence:** 4

**Summary:**

The paper introduces a novel data filtering method named T-MARS (Text Masking and Re-Scoring) tailored for extensive image-text datasets.
An analysis of the LAION dataset revealed that 40% of the images have text overlapping with the associated caption, leading models to rely on OCR instead of learning from visual features which is the motivation for building T-MARS.
The proposed methodology involves detecting text within images, masking it out, and subsequently re-scoring the image in relation to the caption using the CLIP model. Images that score low are discarded, ensuring the retention of images where visual features are still closely correlated with the text.

**Strengths:**

The overall idea it's straightforward and easy to understand.

The paper shows good empirical results. When using the proposed method to filter out data there is an increase in accuracy.

The filtering method was applied on the LAION dataset and the trained models on the newly curated dataset are tested on a decent amount of downstream tasks.

The paper findings are in line with other works (such as [1]) that show that data quality is important.

[1] Gunasekar, Suriya, et al. "Textbooks Are All You Need." arXiv preprint arXiv:2306.11644 (2023).

**Weaknesses:**

The motivation for the work is somehow weak and it lacks theoretical analysis of why text-only images degrade visual learning compared to mislabeled data.

Chapter 3: manually analyzes 500 sample images from the LAION dataset to categorize them based on the correlation between image features (text or visual) and the caption. It lacks some metrics to quantify how representative the 500 sample is of the whole dataset. I appreciate that additional details are given in the appendix, however the work would benefit for more experiments, more details and more analytics. For example, take a larger random sample with statistical estimates of error bars on proportions.

Chapter 6: it is very hard to follow. A rewriting of it to better present the experiments would be beneficial.

The whole method relies on CLIP score for filtering, which can be noisy and introduce additional biases. The current version of the paper is not tackling this.

**Questions:**

What happens if the model is trained with the masked images? So instead of discarding them, you train the model with the masked images.

Are there other datasets that might benefit from the proposed method? (maybe CC12M [1])


[1] Changpinyo, Soravit, et al. "Conceptual 12m: Pushing web-scale image-text pre-training to recognize long-tail visual concepts." Proceedings of the IEEE/CVF Conference on Computer Vision and Pattern Recognition. 2021.

[After rebuttal] The authors addressed my concerns especially regarding mislabeled data vs text-only images. Thus I will raise my score.

---

> ### Author Response · Authors · 2023-11-20
>
> Thank you for your time and valuable feedback. We are happy to see that you appreciated the simplicity of our approach and the promising empirical results on a variety of tasks. We address all of your concerns below:
>
> ### **Why text-only images hurt, but mislabeled data does not?**
> First, we want to clarify that the CLIP similarity metric is already good (not perfect) at capturing mislabeled data. In fact, most image-caption pairs below a CLIP score of 0.2 are already mislabeled or have random captions. They continue to have a low CLIP score even post-masking and are hence still removed by T-MARS. The natural CLIP score does not, however, help remove text-dominant image-caption pairs. As seen in our synthetic experiment in Section 6, mislabeled examples and text-dominant image-caption pairs hurt visual representations almost equally which supports your insight.
>
> However, we also highlight that in Section 4.2, we proposed two competitive data filtering baselines, C-RHO and C-SSFT, that filter out the mislabeled and ambiguous examples. These baselines have been proposed by drawing insights from the long line of work around mislabeled and hard example mining in the supervised classification regime. Further, as shown in Table 1 and Table 2 (see T-MARS $\cap$ C-SSFT row), taking an intersection of data filtered by T-MARS (text-dominated images) and the proposed baselines (say C-SSFT, which removes additional mislabeled samples that CLIP score didn’t capture) gives additive gains. This highlights the importance of filtering out both the text-dominated and mislabeled samples, as you rightly suggested.
>
> ### **Larger pilot study beyond 500 images**
> We agree with the reviewer that a larger scale extension of our pilot study will definitely be helpful for the community. In Appendix C.2, we talk about statistical estimates of error rates for the estimated proportions. Assuming that the sampling is unbiased, 500 samples are enough to estimate the proportions with a 5% margin of error (95% confidence).
> However, to verify whether the sampling itself is representative of the population (the unbiased assumption above), we performed a similar pilot study over 500 additional examples (once again randomly taken from the LAION dataset). The table below gives the estimated and the actual count of data points from the various categories.
>
> |           | Category 1 | Category 2 | Category 3 | Category 4 | Category 5 |
> | --------- | ---------- | ---------- | ---------- | ---------- | ---------- |
> | Estimated (based on previous study) | 25         | 225        | 50         | 100        | 100        |
> | Actual (new 500 examples study)   | 24         | 232        | 53         | 96         | 95         |
>
>
> ### **Will there be gains on CC12M**
> CC12M is a small and already well-curated dataset. Hence we do not think that there will be significant gains using T-MARS or any other baselines for data filtering here.
> However, note that such small datasets cannot be used for training large-scale state-of-the-art CLIP models, which use noisy webscale image-caption datasets like Common Crawl. Our work focuses on building scalable and efficient curation approaches for these webscale datasets.
>
> ### **Training on masked images**
>  This is indeed an interesting question. We chose not to train on masked images to avoid distribution shifts and loss in performance potentially due to the aberrations introduced by masking (although they are indeed quite negligible, see Appendix E). Moreover, there is no reason to train on masked images from Category 5 (text only) because they will be data points where the text and the (now masked) input are uncorrelated. As far as Category 3 is concerned, the uncorrelated texts are already ignored by the model (as also seen in Section 6). Finally, let us come to the most important category in question (Category 4 with both text and visual features). Currently, we retain the unmasked input in such categories (that have both text and visual features). One proposal would be to only retain the visual features in these (as you suggest). We believe that keeping the text features can be helpful for retaining an understanding of certain visual features as well, such as brands and trademark logos (such as the Superman logo) which can be in-part textual, but also have a distinct visual pattern. Our masking algorithm will mask these patterns and potentially hinder the model’s knowledge.
>
> ### **Section 6**
> We acknowledge your feedback about the difficulty of parsing Section 6. **We have rewritten the same** by focussing on the insights and results in the main paper and moving details to the Appendix that were previously hindering clarity in the interest of brevity. We hope the updated version reads better.
>
> ---
>
> We again thank you for raising these interesting questions and hope that the clarification helps you assess the work more positively. We are happy to provide any further clarifications if requested.

---

> ### Author Response · Authors · 2023-11-22
>
> Dear reviewer,
>
> We worked on addressing all of your concerns in the response above. We hope to hear back from you. As a quick summary, in our rebuttal:
>
> - we highlighted that T-MARS already removes most of the mislabeled images and the paper also proposed 2 baselines focused on filtering out mislabeled images. Interestingly, T-MARS + proposed baselines give additive gains, highlighting the importance of removing mislabeled samples, as you also guessed.
>
> - we conducted an additional pilot study, to show that our estimates for different data type proportions are indeed within the error range estimated statistically (i.e. 2-3%).
>
> - we have rewritten Section 6 (on estimating the utility of each data type), focusing on the insights and key takeaways, as requested.
>
> We believe that these points address all the concerns raised and please let us know if there is anything else that you are still concerned about.

---

> ### Author Response · Authors · 2023-11-22
>
> Dear reviewer,
> Since today is the last day of the author-reviewer discussion period, we kindly request you to please take a look at the rebuttal above, which we believe addresses all the concerns raised by you.

---

> > ### Comment · Reviewer_3vr7 · 2023-11-22
> >
> > The rebuttal clarifies my concerns, thus I raised my score. After reading the rebuttal it's a lot more clear that the mislabeled data is also filtered. Please consider slightly re-writing the paper to clearly state that (even though it might feel like stating the obvious).

---

> ### Author Response · Authors · 2023-11-22
>
> We thank the reviewer for going through the rebuttal and raising the score. We will definitely work on incorporating the above suggestion about mislabeled examples in the manuscript.
>
> Authors

---

### Official Review · Reviewer_hNGm · 2023-11-03

**Soundness:** 3 good
**Presentation:** 2 fair
**Contribution:** 3 good
**Rating:** 6
**Confidence:** 2

**Summary:**

## Summary

This paper aims to improve visual representations via a proposed data filtering approach. It is based on an observation that about 40% images contain overlapped text. The experimental results show the performance of the proposed method to some ext

**Strengths:**

## Strengths
1. The motivation of this paper sounds reasonable.

2. Some experimental results look good.

**Weaknesses:**

## Weaknesses
1. The writing of this paper is somewhat obscure, resulting in that it is some difficult to follow this paper.

2. Is it possible to directly remove all the text in the images? This may avoid the distractions of the text.

3. It would be better to conduct experiments on more datasets, except for LAION.

**Questions:**

Please see above.

---

> ### Author Response · Authors · 2023-11-20
>
> Thank you for your time and valuable inputs. We provide clarifications for the concerns raised in your review:
>
> ### **Directly remove all the text in the images**
> The first step of TMARS, i.e. text-detection can indeed be seen as trying to identify and remove all the texts in the images. However, note that TMARS then filters out the images with low CLIP score after masking.
> This is because, as outlined in our pilot study (Section 3), a significant number of images primarily contain text as the sole feature correlating with the caption.  It would be suboptimal (waste of compute) to retain these images and spend training compute on them, as after masking, they do not have any visual features (correlated with caption) that could possibly help the model to learn better visual representations.
>
> ### **Experiments on more datasets**
> We point to Table 2, where we also show results on the DataComp, a latest benchmark for multimodal data filtering. For example, on the medium scale of datacomp, TMARS observes around $6.5\%$ gains on ImageNet zeroshot accuracy compared to CLIP filtering and $4.4\%$ compared to TextMatch[1]. The goal of our work is to improve training on noisy, web-crawled datasets, and in that regard, we have shown effectiveness on both, filtering the common crawl, and pre-filtered LAION datasets. The common crawl is pretty much all the data on the web, and captures any subset of data that may be created out of it. We hope this helps acknowledge the significance of our results.
>
>
> ### **Writing**
> **We have re-written** parts of the paper that we felt might have been hard to follow. This especially included a re-write of Section 6. If you have any particular comments on any other parts of the paper, we would be happy to make amends. Thanks!
>
> ---
>
> [1] Filip Radenovic, Abhimanyu Dubey, Abhishek Kadian, Todor Mihaylov, Simon Vandenhende, Yash Patel, Yi Wen, Vignesh Ramanathan, and Dhruv Mahajan. Filtering, distillation, and hard negatives for vision-language pre-training. arXiv:2301.02280, 2023.

---

> > ### Author Response · Authors · 2023-11-22
> >
> > Dear reviewer,
> >
> > We worked on addressing all of your concerns in the response above. We hope to hear back from you. As a quick summary, in our rebuttal:
> >
> > - we clarified that our experiments are on LAION as well as DataComp, which is the latest benchmark for multimodal data filtering.
> > - we highlighted the suboptimality of just removing the text over all the images (as suggested by the reviewer).
> > - we have rewritten parts of the paper, which we believed were difficult to parse.
> >
> > We believe that these points address all the concerns raised and please let us know if there is anything else that you are still concerned about.

---

### Official Review · Reviewer_ZjEa · 2023-11-05

**Soundness:** 3 good
**Presentation:** 3 good
**Contribution:** 3 good
**Rating:** 6
**Confidence:** 2

**Summary:**

The paper aims at filtering out irrelevant data based on text masking and re-scoring to help the learning of visual features for zero- and few-shot image recognition. The proposed method is simple and can improve the zero-shot performance by only modifying the subset of data and evaluating multiple tasks to show no bias issue in the filtered subset. In addition, the experimental results show that the proposed method brings promising high-quality data curation for data filtering.

**Strengths:**

1. Data cleaning is an important topic in the deep learning field. The proposed filtering data method shows the observation that nearly 40% of LAION’s images contain text overlapping the caption and then designs a method to eliminate the noise for the data filtering.
2. Instead of simply adding or removing data, the authors mask out the text in an image and restore the text regions by replacing the region with the average color of the surrounding pixels. Then, the similarity score between the image and the caption is calculated in order to filter out the low-score images.
3. The proposed method is evaluated on multiple baselines ranging from 2 million to 128 million to demonstrate robustness.

**Weaknesses:**

1. Even though the proposed method has been evaluated on multiple datasets and various tasks, the metric is only the accuracy, which may be narrow and bias may exist for other metrics.
2. The image's text overlaps with the caption which may not be helpful for learning visual features, a subsection for the discussion with multiple ways to resolve the issue can help researchers get more insight into this topic instead of simply exploiting the masking technique.

**Questions:**

1. The proposed filtering method is reasonable, but can this method be used for all different tasks with only one metric, i.e., accuracy? Is that possible that the method filtered some salient signals but isn't shown in this paper due to the single metric?
2. When masking out the text of an image, Will the inpaint technique alter the original data? If it's not an important issue, is that possible to leverage the power of generative models for it?
3. It'll be good if the authors provide the distribution scores (cosine similarity) before and after filtering out the data that can help understand the distribution of the good/bad data.
4. In this paper, the proposed method removes the text of an image, will it be different if using different percent of the masking?
5. After filtering out the data, is data augmentation used in the experiments? Will it provide a more significant improvement?

---

> ### Author Response · Authors · 2023-11-20
>
> Thank you for your time and valuable feedback. We are happy to note that you appreciated the simplicity of our method, supported by promising results. We provide clarifications to your questions below:
>
> ### **Accuracy as the only metric**
> Sorry for not being precise about this, but we follow the standard evaluation metrics as used in the DataComp benchmark (https://github.com/mlfoundations/datacomp/blob/main/tasklist.yml). For example, on retrieval tasks (MSCOCO and Flickr) the metric is mean_recall@1 and on the WinoGAViL dataset it is jaccard_score. Further, we have updated Table 2 in the paper with an additional column of average performance across all the datasets in DataComp, which involves F1 score as the metric for some of the class imbalanced datasets like wilds-iWILDCam and worst region accuracy as the metric for wilds-FMoW.
>
>
> ### **Distribution of scores before and after masking**
> Thank you for the suggestion. Recall our 500-example pilot study where we hand-labeled examples into various categories, like those with only visual features or those with only text features. We calculated the CLIP similarity score of these image-caption pairs, before and after masking the text in the respective images. **A histogram of this distribution of scores (before and after masking) is now uploaded in Appendix F**. Our observations indicate that after masking, the CLIP score of images with just text features (OCR) drops significantly (indicating they will be filtered out by T-MARS). At the same time, CLIP scores of images with only visual features stay nearly the same. This demonstrates the efficacy of the filtering method for removing samples with only text features.
>
> ### **Data augmentation after filtering**
> We do not perform any additional data augmentation after filtering out the bad data to ensure a fair comparison with the baselines. We follow the standard and widely used CLIP training implementation given here[https://github.com/mlfoundations/open_clip]. Indeed, data augmentation may provide additional improvements in visual representation, but our work aims to improve representations by intervening on the data, not the training process.
>
> ### **Does inpainting alter original data?**
> If the text overlays a patch with significant visual features, inpainting might create some aberrations. However, in general, we observe that the text detection algorithm (FAST[3]) gives very tight bounding boxes around the text, leading to no major aberrations of visual features when inpainting the text area, serving our main purpose in this work well. In Appendix E, we have added some pairs of original and masked images to highlight the same.
>
> Additionally, the histogram of CLIP score distribution in Appendix F (which you suggested) provides evidence that the method works well at masking out text features, thereby, lowering CLIP scores of only the images with just text features. However, we agree with you that exploring more nuanced ways to remove the text from the image is an interesting direction for future work.
>
> ### **Impact of percent of Masking**
> To decide the threshold of image-caption pairs to be removed post-masking, we use a CLIP-similarity threshold of 0.281, which is the same as that used in prior work, such as LAION filtering. Fine-tuning the threshold may indeed bring further gains, but running such parameter searches is beyond our compute scope.
> ### **New Discussion: Appendix G (Other ways to resolve text overlap with caption)**
> We have added a discussion section on other ways to circumvent text feature learning in Appendix G  (Page 26 and Page 27), as requested by you.  We give a brief summary here:
> - One can remove all inputs with any text features matching the caption. However, this is suboptimal as it removes datapoints with both visual and text features [1].
> - Alternatively, one can consider a much more expensive process of modifying the input captions of the dataset. One way for the same would be to remove identified text OCR from the input captions. Another way would be to generate new captions altogether, as recently explored in [2].
> - Finally, one could also consider modifying the input images to in-paint the OCR region. Note here that in T-MARS, we chose to use the masked images just to calculate new CLIP scores for filtering. A more elaborate discussion is provided in the appendix.
> ---
>
> We again thank you for raising these interesting questions and hope that the clarification helps you assess the work more positively. We are happy to provide any further clarifications if requested.
>
> [1] https://arxiv.org/abs/2301.02280.
> [2] https://arxiv.org/abs/2307.10350.
> [3] https://github.com/czczup/FAST.

---

> > ### Author Response · Authors · 2023-11-22
> >
> > Dear reviewer,
> >
> > We worked on addressing all of your concerns in the response above. We hope to hear back from you. As a quick summary, in our rebuttal:
> >
> > - we added a distribution of CLIP score before and after masking (as requested), which elucidates that the CLIP score of only the images dominated by text features (OCR) drops. This highlights the efficacy of our approach.
> > - we clarified that accuracy is NOT the only metric considered. Our evaluation spans F1 score, worst region accuracy (for class imbalanced datasets), and mean_recall scores (for retrieval tasks).
> > - we added a discussion on the effect of inpainting on the original data.
> >
> > We believe that these points address all the concerns raised and please let us know if there is anything else that you are still concerned about.

---

> > > ### Author Response · Authors · 2023-11-22
> > >
> > > Dear Reviewer,
> > >
> > > Since today is the last day of the author-reviewer discussion period, we kindly request you to please take a look at the rebuttal above, which we believe addresses the 2 weaknesses and the concerns identified by you. In summary,
> > >
> > > 1. **We wrote a new section Appendix G (Page 26, 27)** providing an elaborate discussion on various ways to circumvent text feature learning.
> > > 2. **We added a new section Appendix F (Page 26)** providing distribution of CLIP scores before and after masking (as requested), which elucidates that the CLIP score of only the images dominated by OCR drops when we mask out text-features in images.
> > > 3. We answered the remaining questions in the first rebuttal response above.
> > >
> > > Thank you for your time with the original review. We hope to hear back from you to see if there is anything else that you may be concerned about.

---

### Meta-Review · Area_Chair_MeB1 · 2023-12-14

**Metareview:**

The paper proposes an algorithm to filter web datasets used to training CLIP, for learning better visual representations and improving zero-shot accuracy. Specifically, for an unlabeled image captioning dataset, the algorithm filters out those pairs where the text dominates the remaining visual features, or those can yield captions only with text in the image. The curated datasets can make the model focus more on visual features.

The experiments are done on several datasets ranging from 2M to 128M samples, and the results show the effectiveness. All reviewers give positive feedbacks to the paper, and the authors have resolved most of their concerns.

Strengths:
1. Data cleaning is an important topic in the deep learning field.
2. The overall idea it's straightforward and easy to understand.
3. The proposed method is simple yet effective.
4. The paper shows good empirical results.

Weaknesses:
1. It lacks theoretical analysis of why text-only images degrade visual learning compared to mislabeled data.
2. There are still some typos. E.g., “why In this section” at the very beginning of Sec. 6.

**Justification For Why Not Higher Score:**

The paper itself is a good one. While it provides limited insights to a future direction except that the data cleaning is significant. I expect an oral/spotlight paper (if a higher score is given) can give more insights to other researchers. Besides, some reviewers also raise their concerns about the motivation of the paper.

**Justification For Why Not Lower Score:**

1. Data cleaning is an important topic in the deep learning field. Without extra human-labeling, the authors clean the dataset through a simple method.
2. The results show the effectiveness of the proposed method.
3. The paper is well-written.
4. The authors have resolved most of the concerns from reviewers, and all reviewers give positive scores to the paper

Totally, I think the paper worth an acceptance.,

---

### Decision · Program_Chairs · 2024-01-16

Accept (poster)